# Estimating Potential Outcome Distributions with Collaborating Causal Networks

**Tianhui Zhou**                                                    *thuizhou@gmail.com*
*Department of Biostatistics and Bioinformatics*
*Duke University*
*Durham, NC 27705, U.S.*

**William E Carson IV**                                      *william.carson@duke.edu*
*Department of Biomedical Engineering*
*Duke University*
*Durham, NC 27705, U.S.*

**David Carlson**                                             *david.carlson@duke.edu*
*Department of Civil and Environmental Engineering*
*Department of Biostatistics and Bioinformatics*
*Department of Computer Science*
*Department of Electrical and Computer Engineering*
*Duke University*
*Durham, NC 27705, U.S.*

**Reviewed on OpenReview:** *https://openreview.net/forum?id=q1Fey9feu7*

## Abstract

Traditional causal inference approaches leverage observational study data to estimate the difference in observed (factual) and unobserved (counterfactual) outcomes for a potential treatment, known as the Conditional Average Treatment Effect (CATE). However, CATE corresponds to the comparison on the first moment alone, and as such may be insufficient in reflecting the full picture of treatment effects. As an alternative, estimating the full potential outcome distributions could provide greater insights. However, existing methods for estimating treatment effect potential outcome distributions often impose restrictive or overly-simplistic assumptions about these distributions. Here, we propose Collaborating Causal Networks (CCN), a novel methodology which goes beyond the estimation of CATE alone by learning the *full potential outcome distributions*. Estimation of outcome distributions via the CCN framework does not require restrictive assumptions of the underlying data generating process (e.g. Gaussian errors). Additionally, our proposed method facilitates estimation of the utility of each possible treatment and permits individual-specific variation through utility functions (e.g. risk tolerance variability). CCN not only extends outcome estimation beyond traditional risk difference, but also enables a more comprehensive decision making process through definition of flexible comparisons. Under assumptions commonly made in the causal inference literature, we show that CCN learns distributions that asymptotically capture the correct potential outcome distributions. Furthermore, we propose an adjustment approach that is empirically effective in alleviating sample imbalance between treatment groups in observational studies. Finally, we evaluate the performance of CCN in multiple experiments on both synthetic and semi-synthetic data. We demonstrate that CCN learns improved distribution estimates compared to existing Bayesian and deep generative methods as well as improved decisions with respects to a variety of utility functions.

# 1  Introduction

Personalized medicine requires estimating how an individual's intrinsic biological characteristics trigger unique and heterogeneous responses to treatments (Yazdani & Boerwinkle, 2015). Under the causal inference potential outcomes framework (Imbens & Rubin, 2015) these treatment effects are characterized by the difference between the conditional expectations of potential outcomes under different treatment assignments, known as the Conditional Average Treatment Effect (CATE[1]). Since only the outcome for the assigned treatment is observed, estimating CATE often requires inferring the unobserved or counterfactual potential outcomes (Ding & Li, 2018). Recently, machine learning approaches have been developed for CATE estimation including extensions of Random Forests (Wager & Athey, 2018) and bespoke neural network frameworks (Shalit et al., 2017; Shi et al., 2019), among other methods.

However, CATE does not necessarily align with optimal choices. In a decision theoretic framework, the optimal decision maximizes the expected utility function $U(\gamma)$ over the distribution of outcomes (Joyce, 1999), where a utility function $U(\gamma)$ quantifies an individual's preferences or tolerances to certain treatments with respects to $\gamma$, the variable over which these preferences or tolerances vary. CATE represents a special case of an identity utility function $U(\gamma) = \gamma$; however, more complex utility functions require alternative estimation approaches. One approach to learning a decision maker is to make the utility function itself the objective function and optimize with respects to transformed outcomes, known as policy learning (Qian & Murphy, 2011; Kallus & Zhou, 2018). However, traditional policy learning methods require the utility function to be pre-specified, whereas utility functions typically vary between individuals (Pennings & Smidts, 2003). Defining proper losses with respects to complex decision criteria could be challenging from both a theoretical as well as a computational standpoint. Moreover, if we also want to account for an individual's specific needs or have a trained model generalize to new individuals with different utilities, traditional policy learning methods fall short. Previous efforts to estimate potential outcome distributions include Bayesian Additive Regression Trees (BART) (Chipman et al., 2010; Hill, 2011), variational methods (Louizos et al., 2017), generalized additive models with location, scale and shape (GAMLSS) (Hohberg et al., 2020), and techniques based on adversarial networks (Yoon et al., 2018; Ge et al., 2020). These techniques often impose explicit or implicit assumptions about the outcome distributions (e.g. Gaussian errors) which may not align with the true data generating mechanism.

To address these issues, we propose a novel neural network-based approach, Collaborating Causal Networks (CCN), to estimate the full potential outcome distributions, in turn providing flexible personalization and valuable insights with regards to specific individuals. CCN extends the structure of the Collaborating Networks (CN) framework (Zhou et al., 2021) to create a new causal framework that flexibly represents distributions. Under assumptions commonly made in the causal inference literature, we show that CCN asymptotically captures conditional potential outcome distributions without having to explicitly make restrictive assumptions about the form of these distributions. We also introduce an adjustment method that alleviates effects of imbalance between treatment groups, thus addressing a common confound that impairs model generalization. Empirically, this adjustment method improves point estimates, distribution estimates, and decision-making. We summarize the contributions of our work as follows:

1. We propose a novel framework, Collaborating Causal Networks (CCN), to estimate full potential outcome distributions.

2. We characterize the asymptotic properties of CCN for estimation of smooth outcome distributions (i.e., distributions without point masses) by extending CN theory to the causal inference setting.

3. We propose an adjustment scheme that combines domain-invariant, propensity-specific information and propensity stratification to alleviate the effects of treatment group imbalance.

4. We evaluate our framework with respect to different personalized utility functions in causal inference decision making, thus demonstrating the ability of CCN to address distinct user needs when such personalized utilities are made available.

5. We demonstrate empirically that CCN improves conditional decisions in a potential outcomes framework.

---

[1]CATE is sometimes also referred to as the Individual Treatment Effect, or ITE (Shalit et al., 2017; Yao et al., 2018).

## 2 Problem Statement

### 2.1 Notations

Let $X$ denote the random variable from which observed covariates $x$ are drawn, $X \in \mathcal{X} \subset \mathbb{R}^p$. We assume a binary treatment condition with each unit or observation assigned a treatment $T \in \{0, 1\}$. We choose the binary setup for clarity; however, the extension to multi-class problems is straightforward to construct under the proposed framework. We let $Y(0) \in \mathbb{R}$ and $Y(1) \in \mathbb{R}$ represent the continuous potential outcomes under the two treatment conditions, with $Y(T)$ representing the observed outcome. Lowercase letters with subscript $i$ are used to denote individual observations on each subject: $\{y_i(1), y_i(0), t_i, y_i(t_i), x_i\}$.

A common goal in the causal inference literature is CATE estimation, $\tau(x_i) = \mathbb{E}[Y(1)|X = x_i] - \mathbb{E}[Y(0)|X = x_i]$. Due to the presence of unmeasured features in practice, Vegetabile (2021) points out that $\tau(x_i)$ is an average taken over individuals with the same observed features. The main objective of this work is to address the problem of conditional causal inference and to develop a framework that addresses a broader range of objectives beyond CATE alone. Specifically, our goal is to use incomplete data to infer the *full distributions* over *both* potential outcomes in a binary treatment scenario, $Y(0)|X$ and $Y(1)|X$. Successful estimation of these distributions will facilitate exploration of personalized needs and preferences through the introduction of various utility functions $U(\gamma)$ (Dehejia, 2005).

### 2.2 Utility Functions

While estimating potential outcome distributions and capturing uncertainties can be helpful to understand predictions in and of themselves, we also want to evaluate whether capturing these distributions leads to improved downstream decisions. We incorporate utility functions as part of our proposed CCN framework as a way to facilitate personalized decisions and recommendations. Utility functions aim to quantify the value of assigning a treatment to a patient based on traits or characteristics of said patient, while also taking into account personal preferences, tolerances, and needs. The structure of utility functions can be adapted to different levels and degrees of comparisons. Below, we present different setups for quantifying utility with the following equations, each providing a different scope of comparisons:

(i) unified utility: $\mathbb{E}_{\gamma \sim p(y(1)|x)}[U(\gamma)] - \mathbb{E}_{\gamma \sim p(y(0)|x)}[U(\gamma)]$;

(ii) treatment-specific utility: $\mathbb{E}_{\gamma \sim p(y(1)|x)}[U_1(\gamma)] - \mathbb{E}_{\gamma \sim p(y(0)|x)}[U_0(\gamma)]$;

(iii) utility related to inherent features: $\mathbb{E}_{\gamma \sim p(y(1)|x)}[U_1(\gamma, x)] - \mathbb{E}_{\gamma \sim p(y(0)|x)}[U_0(\gamma, x)]$;

(iv) utility for personalized needs: $\mathbb{E}_{\gamma \sim p(y(1)|x)}[U_{1,i}(\gamma, x)] - \mathbb{E}_{\gamma \sim p(y(0)|x)}[U_{0,i}(\gamma, x)]$.

Going in order from (i) to (iv)[2] the scope of comparison is broadened. A major distinction between (iii) and (iv) is that, for subjects with same features $x$, (iv) allows the utility to differ in accordance to personal preferences, which better aligns with the concept of personalized medicine. We note that setting $U(\gamma) = \gamma$ in (i) results in the objective for CATE. Therefore, we view the incorporation of utility as a more general framework.

Rather than estimating the potential outcomes and then calculating the utility, a model could be learned to predict the outcome of the utility function in scopes (i), (ii), and (iii). This is the strategy taken in "Policy Learning" (Athey & Wager, 2021). However, transforming the outcomes may result in loss of information. For example, consider a threshold utility, $U(\gamma) = 1_{\gamma > C}$. Applying the transformation $\mathbb{1}[\gamma > C]$ would transform continuous information into binary information, thus resulting in loss of information. On the contrary, estimating distributions not only enables us to study personalized utilities like (iv) but also retain all information in the system during training. Using estimated distributions to evaluate utility functions can be viewed as a two-step procedure or a plug-in estimator, which might be less efficient than direct optimization in some cases (Bickel & Ritov, 2003). However, its flexibility and adaptability in studying various types of utilities makes it advantageous for practical purposes (Grunewalder, 2018).

---

[2](iv) may or may not depend on $X$.

In short, utility functions provide a way to flexibly take into account individual features, needs, and preferences when considering a possible treatment. Thus, the incorporation of utility functions with the CCN forms a framework that enables comprehensive treatment comparisons according to the above considerations.

### 2.3 Causal Assumptions

Like many causal methods, CCN relies on the assumptions of strong ignorability and consistency to estimate the potential outcome distributions when a single treatment outcome is only observed for each datum (Rosenbaum & Rubin, 1983; Hernan & Robins, 2020). Strong ignorability and consistency are characterized via the following assumptions:

**Assumption 1** (Positivity or overlap). *$\forall X \in \mathcal{X} \subset \mathbb{R}^p$, the probability of assignment to any treatment group is bounded away from zero: $0 < \Pr(T = 1|X = x) < 1$, $\forall x$ such that $p(x) > 0$.*

**Assumption 2** (Consistency). *The observed outcome given a specific treatment is equal to its potential outcome: $Y|T, X = Y(T)|T, X$.*

**Assumption 3** (Ignorability or unconfoundedness). *The potential outcomes are jointly independent of the treatment assignment conditional on $X$: $[Y(0), Y(1)] \perp T|X$.*

## 3 Collaborating Causal Networks

The CCN approach approximates the conditional distributions of $Y(0)|X$ and $Y(1)|X$. CCN uses a two-function framework based on the Collaborating Networks (CN) method (Zhou et al., 2021). We choose to extend CN to the causal setting because it automatically adapts to different distribution families, including non-Gaussian distributions, and we believe that this flexibility and robustness will result in more reliable estimates of the different utilities detailed in Section 2.2. We first give an overview of CN, then present the CCN framework in detail, and finally introduce our adjustment strategies. Proofs of all theoretical claims are provided in Appendix A.

### 3.1 Overview of Collaborating Networks

Similar to Generative Adversarial Networks (GAN) (Goodfellow et al., 2014), CN is based on jointly learning two functions, both of which are approximated by neural networks. However, unlike GAN where the two networks have opposing objective functions that force the networks to "compete" against one another, in CN the two networks form a *collaborative* approach in that they work towards the same goal from different angles. Specifically, CN estimates the conditional distribution, $Y|X$, using two neural networks: a network $g(y, x)$ to approximate the conditional CDF, $\Pr(Y < y|X)$, and a network $f(q, x)$ to approximate its inverse. Information sharing is enforced by the fact that the CDF and its inverse are an identity mapping for any quantile $q$: $g(f(q, x), x) = q$. The networks form a collaborative scheme with their respective losses,

$$\text{g-loss} : \mathbb{E}_{q,y,x} \left[ \ell(1_{y < f(q,x)}, g(f(q, x), x)) \right], \tag{1}$$

$$\text{f-loss} : \mathbb{E}_{q,x} \left[ (q - g(f(q, x), x))^2 \right]. \tag{2}$$

The quantile $q$ is randomly sampled (e.g., $q \sim \text{Unif}(0, 1)$). $\ell(\cdot, \cdot)$ represents the binary cross-entropy loss. The parameters of $f$ and $g$ are only updated according to their respective losses. When the objective is the simultaneous minimization of equations 1 and 2, a fixed point of the optimization is at the true conditional CDF and its inverse (Zhou et al., 2021). In this framework, $g(\cdot)$, the function used to approximate the conditional CDF, is considered the main function. $f(\cdot)$ is viewed as an auxiliary function whose job is to extensively search the full outcome space to help $g(\cdot)$ acquire information about the distribution function over the full relevant space. On the other hand, the optimality of $f(\cdot)$ depends on an optimal $g(\cdot)$, as it acquires information solely through inverting $g(\cdot)$. Zhou et al. (2021) show that $f(\cdot)$ can be replaced by other space searching tools, including prefixed uniform distributions, which may result in a minor performance loss in favor of ease of optimization. This ease of optimization can be especially beneficial when combining CN with other regularization terms. We show a depiction of the CN framework from Zhou et al. (2021) in Figure 1.

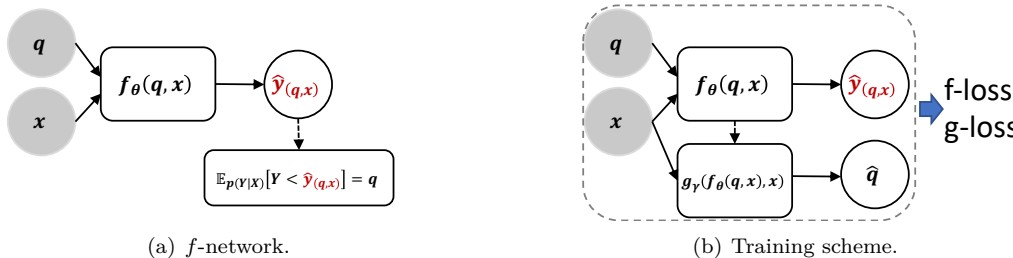

(a) $f$-network.

(b) Training scheme.

Figure 1: Diagram of the CN framework. 1(a) depicts training for prediction of a conditional quantile $\hat{y}(q, x)$ directly. The dashed arrow coming from the component representative of $\hat{y}(q, x)$ indicates that the objective function does not produce a useful gradient. 1(b) shows the full CN framework, where $g(\cdot)$ and $f(\cdot)$ are trained jointly to learn the CDF and conditional CDF.

In this work, we focus on extending the function $g(\cdot)$ and the g-loss for causal inference. We replace $f(q, x)$ with a variable $z$ as a general form of a space searching tool, such as a uniform distribution covering the range of the observed outcomes. We thus simplify the g-loss to,

$$\text{g-loss} : \mathbb{E}_{y,x,z} \left[ \ell(1_{y<z}, g(z, x)) \right]. \tag{3}$$

Thus, equation 3 is a generalization of the g-loss introduced in Zhou et al. (2021). In practice, equation 3 is replaced with an empirical approximation.

### 3.2 Collaborating Causal Networks Formulation

Following the taxonomy of Künzel et al. (2019), CN can be extended either as an "S-learner," where the treatment label is included as an additional covariate and is thus more scalable for multiple treatment groups, or as a "T-learner," where the outcome under each treatment arm is estimated using separate functions or networks. Below, we detail the T-learner extension of the CN. Based on equation 3, we define two functions parameterized by neural networks, $g_0(\cdot)$ and $g_1(\cdot)$, which approximate the untreated group conditional CDF, $\Pr(Y < y | X, T = 0)$, and the treated group conditional CDF, $\Pr(Y < y | X, T = 1)$, respectively. Combining the losses of these two treatment groups results in the following loss function:

$$\text{g-loss}^* = \mathbb{E}_{y(t=0),x,z} \left[ \ell(1_{y(t=0)<z}, g_0(z, x)) \right] + \mathbb{E}_{y(t=1),x,z} \left[ \ell(1_{y(t=1)<z}, g_1(z, x)) \right]. \tag{4}$$

We refer to the optimization of this objective as the CCN framework. Under Assumptions 1, 2, and 3, the fixed point solution and consistency of CCN framework still hold regardless of the treatment group imbalance. To summarize, Assumption 2 connects the conditional distribution $Y | X, T$ to the potential outcome distribution $Y(T) | X, T$ in each treatment space with density functions: $p(x | T = 0)$ and $p(x | T = 1)$. Assumption 1 guarantees that despite the treatment group imbalance, $p(x | T = 0) \neq p(x | T = 1) \neq p(x)$, each treatment space can still have sufficient coverage of the full space $p(x)$ given enough samples. Lastly, Assumption 3 generalizes the potential outcome distributions from each space $Y(T) | X, T$ to the full space $Y(T) | X$ removing its conditioning on the treatment label $T$. Given our assumptions, we state:

**Proposition 1** (Optimal solution for $g_0$ and $g_1$). *When the support of the outcomes is a subset of the support of $z$, or as $z$ covers the whole outcome space, the functions $g_0$ and $g_1$ that minimize g-loss\* are optimal when they are equivalent to the conditional CDF of $Y(0) | X = x$ and $Y(1) | X = x$, $\forall x$ such that $p(x) > 0$.*

**Proposition 2** (Consistency of $g_0$ and $g_1$). *Assume the ground truth CDF functions for $T \in \{0, 1\}$ are Lipschitz continuous with respect to both the features $X$ and the potential outcomes $\{Y(0), Y(1)\}$ and the support of the outcomes is a subset of the support of $z$. Denote the ground truth functions as $g_0^*$ and $g_1^*$. As $n \to \infty$, the finite sample estimators $g_0^n$ and $g_1^n$ have the following consistency property: $d(g_0^n, g_0^*) \to_P 0; d(g_1^n, g_1^*) \to_P 0$ under some metrics $d$, such as the $\mathbb{L}_1$ norm.*

Taken together, these propositions state that the CDF estimators $g_0$ and $g_1$ inherit large sample properties when estimating potential outcome distributions under the CCN framework. Proofs are provided in Appendix

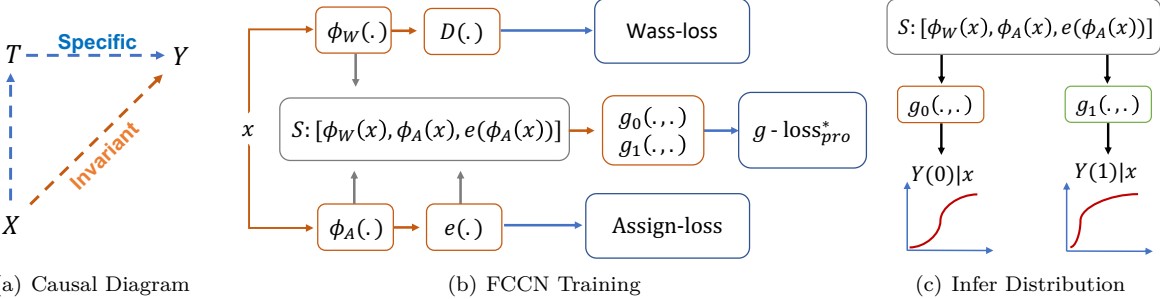

(a) Causal Diagram          (b) FCCN Training          (c) Infer Distribution

Figure 2: 2(a) depicts how two sources of information could impact $Y$. 2(b) visualizes the FCCN network. 2(c) depicts how the trained $g_0$ and $g_1$ functions can be used to sketch the underlying CDFs of $Y(0)|X = x$ and $Y(1)|X = x$.

A. Though consistency is only claimed for functions belonging to the Lipschitz family, we consider this to be a reasonable assumption as Lipschitz functions can provide good approximations to a wide range of functions (Sohrab, 2003), especially considering the boundedness of CDFs between $(0, 1)$. In practice, our approximation functions are implemented as neural networks, which allows for additional flexibility. Enforcing a Lipschitz constraint on a neural network can be done through a variety of techniques, such as by limiting the weights to a finite value via weight clipping (Arjovsky et al., 2017). In practice, the learned functions were smooth and we did not find it necessary to enforce such a constraint.

### 3.3 Adjustment for Treatment Group Imbalance

One obstacle in observational studies is treatment group imbalance, where the distributions of the covariate spaces $p(x|T = 0)$ and $p(x|T = 1)$ differ. This creates two major issues for causal predictions: poor generalization over different treatment spaces and confounding effects. In the asymptotic regime (Proposition 2), this imbalance is less problematic since overlap (Assumption 1) ensures all regions with positive density will eventually be densely covered with samples. However, for a finite number of samples this imbalance hurts inference. Therefore, we pair our method with an adjustment scheme to address this obstacle.

We encode the covariates into a space that simultaneously facilitates generalization between spaces and adjusts for confounding effects. We represent this new space as $S = [\phi_W(x), \phi_A(x), e(\phi_A(x))]$. Two neural networks, $\phi_W(\cdot) : \mathbb{R}^p \to \mathbb{R}^{q_W}$ and $\phi_A(\cdot) : \mathbb{R}^p \to \mathbb{R}^{q_A}$, transform the input $x \in \mathbb{R}^p$ into $q_W$ and $q_A$-dimensional latent spaces. In Figure 2(a), these spaces are graphically depicted as the two sources of information that can impact the outcome. $\phi_W(\cdot)$ is enforced to be invariant to treatment groups (domain-invariant) and the $\phi_A(\cdot)$ is enforced to learn specific differences between the treatment spaces (domain-specific) (Shalit et al., 2017; Ben-David et al., 2010). The invariant component $\phi_W(\cdot)$ finds a more balanced representation between spaces that benefits generalization, while the component $\phi_A(\cdot)$ controls for confounding effects through learning the treatment assignment mechanism. We denote the g-loss$^*$ trained on this representation space as g-loss$^*_{pro}$. Additionally, a neural network $e(\cdot) : \mathbb{R}^{q_A} \to [0, 1]$ uses the output of $\phi_A(\cdot)$ to predict the propensity of treatment assignment, $\Pr(T = 1|X = x)$. While $e(\phi_A(x))$ may seem redundant given $\phi_A(x)$, including $e(\phi_A(x))$ in the covariate space encourages *propensity score stratification* (Hahn et al., 2020).

Domain-invariance is encouraged in the space $\phi_W(\cdot)$ through a penalty on the Wasserstein distance (Wass-loss) between the two treatment arms, and $[\phi_A(\cdot), e(\cdot)]$ is encouraged through a cross-entropy loss on the assigned treatment labels (Assign-loss), as detailed in Sections 3.3.1 and 3.3.2, respectively. The representations $[\phi_W(\cdot), \phi_A(\cdot), e(\cdot))]$ with their respective losses are incorporated as regularization terms. This framework is depicted in Figure 2 and the full loss can be expressed as the sum of g-loss$^*_{pro}$,

$$L(g_0, g_1, \phi_W, \phi_A, e) = \text{g-loss}^*_{pro}(g_0, g_1, \phi_W, \phi_A, e) + \alpha \text{Wass-loss}(\phi_W) + \beta \text{Assign-loss}(\phi_A, e). \qquad (5)$$

The hyperparameters $\alpha$ and $\beta$ adjust the relative importance of their respective losses during learning. Empirically, we have found performance to be robust to selection of these hyperparameter values. We call this full adjustment CCN (FCCN). In summary, in FCCN two additional functions, $\phi_W(\cdot)$ and $\phi_A(\cdot)$, are learned

as extensions of the existing CCN framework to learn domain-invariant and domain-specific representations of the treatment spaces, respectively. These functions are parameterized by neural networks in practice.

### 3.3.1 Wass-loss to Alleviate the Covariate Space Imbalance (Domain-Invariant)

The introduction of Wass-loss is motivated by CounterFactual Regression (CFR) implemented with the Wasserstein distance (Shalit et al., 2017). CFR is a causal estimator based on representation learning $\phi_W(\cdot) : \mathbb{R}^p \to \mathbb{R}^{q_W}$. The goal is to find latent representations where $p(\phi_W(x)|T = 1)$ and $p(\phi_W(x)|T = 0)$ are more balanced or domain-invariant than the original space. We use the Wasserstein-1 distance, which represents the total "work" required to transform one distribution into another (Vallender, 1974). Through the Kantorovich-Rubinstein duality (Villani, 2008), this distribution distance is,

$$W\left(\mathbb{P}_a, \mathbb{P}_b\right) = \sup_{\|D\|_L \le 1} \mathbb{E}_{x \sim \mathbb{P}_a}[D(x)] - \mathbb{E}_{x \sim \mathbb{P}_b}[D(x)].$$

$\|D\|_L \le 1$ represents the family of 1-Lipschitz functions. We approximate this distance by adopting the approach of Arjovsky et al. (2017). This turns into the following regularization term,

$$\text{Wass-loss} : \max_D \mathbb{E}_{x \sim p(x|t=1)}[D(\phi_W(x)] - \mathbb{E}_{x \sim p(x|t=0)}[D(\phi_W(x)].$$

$D(\cdot)$ is parameterized by a small neural network, and the Lipschitz constraint on $D$ is enforced through weight clipping. The Wass-loss penalizes differences in the latent space between the treatment and control group, which in turn improves generalization between groups.

### 3.3.2 Assign-loss and Propensity Stratification for Confounding Effects (Domain-Specific)

The introduction of Assign-loss is inspired by Dragonnet (Shi et al., 2019), a deep learning method that learns a latent representation for treatment assignment mechanism to control for confounding effects. It is defined as a binary cross-entropy loss on prediction of the treatment assignment label with $e(\phi_A(x))$,

$$\text{Assign-loss} : -\mathbb{E}_{t,x}\left[t \log(e(\phi_A(x)) + (1 - t) \log(1 - e(\phi_A(x))\right].$$

We additionally incorporate the estimated propensity $e(\phi_A(x))$ directly into our covariate space, an adjustment we term propensity stratification (PS). Using propensity scores in the predictive model is a form of continuous stratification to reduce the bias of estimation by facilitating information sharing within sample strata created by $e(\phi_A(x))$ (Hahn et al., 2020). PS corresponds to learning propensity scores via a cross-entropy objective when estimated propensity scores are included in the original feature space $x$. As a second step, this new feature space $(x, e(x))$ is used to train functions $g0(\cdot)$ and $g1(\cdot)$.

## 4 Related Work

**CATE Estimation.** A common approach to CATE estimation with machine learning is matching, which identifies pairs of similar individuals (Rubin, 1973; Rosenbaum & Rubin, 1983; Li & Fu, 2017; Schwab et al., 2018). This idea motivates many tree-based methods that identify similar individuals within automatically-identified regions of the covariate space (Liaw & Wiener, 2002; Zhang & Lu, 2012; Athey & Imbens, 2016; Wager & Athey, 2018). Deep learning methods have also been proposed for CATE prediction. These include networks with additional loss terms to encourage a treatment-invariant space (Johansson et al., 2020; 2016; Du et al., 2021) and networks that explicitly encode treatment propensity information (Shi et al., 2019). Representation learning can be combined with weighting strategies to enforce covariate balance (Assaad et al., 2021; Hassanpour & Greiner, 2019; 2020a). Despite some resemblance to previous methods in learning more balanced feature embeddings, this manuscript performs an extensive ablation study to support and justify that each component of our proposed method leads to increased robustness. Additionally, these methods largely focus on CATE estimation only (with some exceptions noted below), which may be insufficient to reflect the full picture of different treatment regimes (Park et al., 2021). In contrast, CCN estimates full distributions to assess the utility and confidence of a decision.

**Potential Outcome Distribution Sketching.** Multiple Bayesian methods have been proposed for the estimation of outcome distributions, including Gaussian Processes (Alaa & van der Schaar, 2017), Bayesian dropout (Alaa et al., 2017), and Bayesian Additive Regression Trees (BART) (Chipman et al., 2010). BART has gained popularity in recent years and has been the focus of further modifications, including variations to account for regions with poor overlap (Hahn et al., 2020). However, Bayesian methods can suffer under model misspecification (Walker, 2013), such as mismatch between the assumed and true outcome distributions. Bayesian methods have also been integrated with deep learning, such as the Causal Effect Variational Autoencoder (CEVAE) and its extensions (Louizos et al., 2017; Jesson et al., 2020); hybrid architectures are sometimes adopted to account for certain types of missing data mechanisms (Hassanpour & Greiner, 2020b).

Frequentist approaches can achieve flexible representations of distributions. A well-known adaptation is the Generalized Additive Model with Location, Scale and Shape (GAMLSS), which estimates the parameters for a baseline distribution with up to three transformations given a specific distribution family (Briseño Sanchez et al., 2020; Hohberg et al., 2020). The CDF may also be estimated nonparametrically by adapting density estimation methods such as nearest neighbors (Shen, 2019), which is less reliable in areas of treatment group imbalance. GAN-inspired methods, including GANITE (Yoon et al., 2018), can also learn non-Gaussian outcome distributions. There is emerging literature on conformal prediction in treatment effect estimation (Chernozhukov et al., 2021; Lei & Candès, 2021). However, conformal prediction only learns a specific level of coverage and coverage probabilities are proven for marginal rather than conditional distributions.

**Quantile Regression.** Quantile regression is an established method to estimate uncertainty and variability of continuous outcomes (Koenker & Hallock, 2001; Meinshausen & Ridgeway, 2006; Tagasovska & Lopez-Paz, 2019). Aspects of CN and the proposed CCN framework both bear a resemblance to quantile regression, with the $f$ network described in Section 3.1 learning a quantile regression function. However, the learning methodology is different: canonical quantile regression uses a tilted absolute value loss function, whereas CN focuses on learning the conditional CDF, $g(\cdot)$, and uses different loss functions. These modifications are necessary when using deep networks, as applying the tilted absolute value loss function in a deep network can result in drastically underestimated variability (Zhou et al., 2021). Quantile regression has been studied in the context of causal inference (Chernozhukov & Hansen, 2006; Tagasovska et al., 2020; Sun et al., 2021). However, these works looked at the *population-level* effects of endogenous variables/treatments on potential outcomes rather than individual-level effects and do not consider incorporation of personalized utility functions. Furthermore, many of these studies are not as flexible as the proposed approach and would not work with more complex estimators, in contrast to the deep learning approach proposed here. The comparisons made in the original CN work suggest that the CN approach is quite robust compared to quantile regression losses (Zhou et al., 2021). Regardless, quantile regression can still be beneficial when simpler estimator forms hold or when learning is data-limited.

**Policy Learning and Utility Functions.** A key purpose of estimating the conditional causal effect is to serve as information in decision making processes. A common strategy is policy learning, where the policy is expressed as a function of the feature space and learn a policy that optimizes a pre-defined utility (Beygelzimer & Langford, 2009; Qian & Murphy, 2011; Bertsimas et al., 2017; Kallus & Zhou, 2018). This involves transforming observed outcomes in accordance to the pre-defined utility as the new objective for optimization. Traditionally, utilities studied in policy learning are linear transformation of the potential outcomes, which can be described as the difference between the benefit and cost (Athey & Wager, 2021). However, when we are presented with threshold based utilities, transforming the observed outcomes may be subject to information loss (e.g., binarization of a continuous variable greatly reduces information) according to the Data Processing Inequality (Beaudry & Renner, 2012). Additionally, policy learning can only study pre-specified utilities, whereas a trained potential outcome distribution estimator can serve as a one-stop shop to explore a myriad of personal utility functions.

**Off-Policy Evaluation.** Estimating utility differences and choosing optimal decisions is closely related to the concept of off-policy evaluation (Thomas & Brunskill, 2016). We note that the policy learning strategies above could be considered as off-policy evaluations as they evaluate decisions with respects to utility functions. In reinforcement learning, off-policy evaluation is frequently used to estimate the reward, which is analogous to the utilities examined in this paper. Off-policy evaluation methods have been developed to estimate distributional properties and uncertainties over the return (cumulative reward), including first estimating

the confidence interval on the estimator (Thomas et al., 2015). These methods have been recently extended to develop estimators for the complete distribution of the return through Universal Off-Policy Evaluation (Chandak et al., 2021). However, these methods differ from the proposed approach in the overall goal, as off-policy evaluation estimates the *overall* return and is more similar to estimating the average distribution of difference of utilities, allowing off-policy evaluation methods to take advantage of repeated observations for their theoretical proofs. In contrast, CCN estimates potential outcome distributions for individual data samples, allowing for more personalized estimates and treatment recommendations.

## 5 Experiments

We follow established causal literature and use semi-synthetic scenarios to assess estimates of conditional causal effects. First, we evaluate causal methods using the Infant Health and Development Program (IHDP) dataset (Hill, 2011), where the outcome of each subject is simulated under a standard Gaussian distribution with a heterogeneous treatment effect. The IHDP dataset represents an ideal scenario for methods with Gaussian assumptions, including BART and CMGP. The second dataset used is derived from data collected in a field experiment in India studying the impact of education (EDU). In this case, we synthesize each individual outcome with heterogeneous effect and variability using a non-Gaussian distribution. The semi-synthetic procedures are briefly outlined in Section 5.3 with full details in Appendix B. Additionally, we provide evaluations on a number of different synthetic outcome distributions to compare methods under different scenarios, including multi-modal distributions and several other different distribution families.

We include our base approach, CCN, and its adjusted version, FCCN, in experiments. We compare them to existing approaches that estimate potential outcome distributions including Bayesian approaches (BART (Hill, 2011), CMGP (Alaa & van der Schaar, 2017), CEVAE (Louizos et al., 2017)), a frequentist approach (GAMLSS (Hohberg et al., 2020)), and a GAN-based approach (GANITE (Yoon et al., 2018)). Causal Forests (CF) (Wager & Athey, 2018) is benchmarked for non-distribution metrics as a popular recent CATE-only method. GAMLSS's flexibility and strength in estimating distributions is dependent on a close match to the true distribution families from which data is generated, which is rarely known in practice. Therefore, we evaluate GAMLSS by providing the closest possible distribution to the distributions underlying data generation, meaning that GAMLSS is provided *more information than any other method*. We also benchmark the proposed approaches against policy learning approaches on decision-making metrics. To fully understand the impact of the various adjustments in FCCN compared to CCN, we run ablation studies and evaluate the performance over a suite of hyperparameters for tuning the adjustment.

Details for all model implementations, including model architectures and hyperparameter selection procedures, are provided in Appendix C. We use a neural network-based architecture for CCN and FCCN, but also detail an alternative structure that enforces a monotonic constraint in Appendix D. Model and experiment code is available at `https://github.com/carlson-lab/collaborating-causal-networks`.

### 5.1 Metrics

We evaluate mean outcome estimates via the Precision in Estimation of Heterogeneous Effect (PEHE) metric and the full distribution by estimating the log-likelihood (LL) of the potential outcomes. We regard LL as the key evaluation metric since it evaluates estimation of full distributions. In addition, we evaluate how well each method makes decisions by using Area Under the Receiver Operating Characteristic Curve (AUC) for chosen utility functions to show that improved distributional estimates lead to improved decisions.

**Precision in Estimation of Heterogeneous Effect (PEHE):** We adopt the definition of PEHE defined in Hill (2011). For unit $i$ and covariates $x_i$, we have the CATE and its estimates as $\tau(x_i) = E[Y(1)|X = x_i] - E[Y(0)|X = x_i]$ and $\hat{\tau}(x_i)$. PEHE quantifies the distance between CATE and estimates according to the following:

$$\text{PEHE} = \sqrt{\frac{1}{N} \sum_{i=1}^{N} [(\hat{\tau}(x_i) - \tau(x_i)]^2} \ .$$

Table 1: Quantitative results on IHDP (Section 5.2). The mean and standard error of each metric are reported. CMGP attains top marks in all metrics due to modelling assumptions that align with the true data generating mechanism. FCCN outperforms CCN in all metrics with statistical significance, and remains competitive with CMGP on LL and both AUC metrics. GANITE is only used to estimate the CATE as it is challenging to train on this small dataset according to Yoon et al. (2018).

| Metrics | CCN | FCCN | GAMLSS | GANITE | BART | CMGP | CEVAE | CF |
|---|---|---|---|---|---|---|---|---|
| PEHE | $1.59 \pm .16$ | $1.13 \pm .14$ | $3.00 \pm .39$ | $2.40 \pm .40$ | $2.23 \pm .33$ | $\mathbf{.714 \pm .087}$ | $2.60 \pm .10$ | $3.52 \pm .57$ |
| LL | $-1.78 \pm .02$ | $-1.62 \pm .02$ | $-2.34 \pm .13$ | * | $-1.99 \pm .08$ | $\mathbf{-1.51 \pm .01}$ | $-2.82 \pm .08$ | NA |
| AUC (linear) | $.925 \pm .011$ | $.942 \pm .010$ | $.930 \pm .010$ | $.723 \pm .017$ | $.923 \pm .009$ | $\mathbf{.957 \pm .012}$ | $.523 \pm .008$ | $.896 \pm .009$ |
| AUC (threshold) | $.913 \pm .011$ | $.935 \pm .010$ | $.925 \pm .010$ | * | $.917 \pm .009$ | $\mathbf{.955 \pm .012}$ | $.564 \pm .010$ | NA |

**Log Likelihood (LL):** Log likelihood (LL) measures how well each method models potential outcome distributions. Typically, LL is calculated by evaluating the probability density function (PDF) functions at the observed points. However, closed-form distributions are not directly available for sampling-based approaches such as GANITE and variants of CCN. Instead, we approximate LL by calculating the probability on a neighborhood of the realized outcome $y$, $B_{y,\epsilon} = (y - \epsilon, y + \epsilon)$, where $\epsilon$ is a small positive value:

$$\text{LL} = \frac{1}{2N} \sum_{t=0}^{1} \sum_{i=1}^{N} \log(\hat{Pr}[Y_i(t) \in B_{y_i(t),\epsilon} | X_i = x_i]).$$

Asymptotically, the true distribution dominates in this evaluation, and this can be shown under the criterion of the Kullback–Leibler divergence (Kullback & Leibler, 1951) as $N \to \infty$ and $\epsilon \to 0$. We define $\epsilon = 0.5$ for IHDP and $\epsilon = 0.2$ for EDU to adjust for the scale of the outcomes.

**Area Under the Curve (AUC):** As mentioned in 2.2, personalized decisions can be described by the quantity: $\tau(x_i, U_{0,i}, U_{1,i}) = \mathbb{E}_{\gamma \sim p(y(1)|X=x_i)}[U_{1,i}(\gamma, x)] - \mathbb{E}_{\gamma \sim p(y(0)|X=x_i)}[U_{0,i}(\gamma, x)]$. The optimal decision is based on the sign of this contrast $1_{\tau(x_i, U_{0,i}, U_{1,i})}$, which is regarded as the true label. Using the estimated contrast $\hat{\tau}(x_i, U_{0,i}, U_{1,i})$ as the decision score, we can estimate the AUC by varying the decision threshold.

## 5.2 IHDP

The Infant Health and Development Program (IHDP) dataset is derived from results of a randomized experiment, and the data is transformed to be more observational study-like by removing a non-random portion of the data belonging to the treatment group. We use response surface B as described in Hill (2011) for estimating heterogeneous treatment effects. The study consists of 747 subjects (139 in the treated group) with 19 binary and 6 continuous variables ($x_i \in \mathbb{R}^{25}$). We use 100 replications of the data for out-of-sample evaluation by following the simulation process of Shalit et al. (2017).

The quantitative results are summarized in Table 1. CMGP achieves top marks, as its Gaussian noise model aligns with the true data generating mechanism in this dataset resulting in a significant advantage given limited data. This contrasts to later experiments when its strong assumptions are invalid. Overall, CCN is either competitive with or outright outperforms other methods in estimating both mean and distribution metrics. The advantages of combining the proposed adjustment strategy is evident as FCCN improves over CCN by a clear margin. It is worth noting that LL calculated under the ground truth model is -1.41, demonstrating that FCCN is also effective in capturing the true distributions with an LL estimate of $-1.62 \pm 0.02$ that is only behind the LL estimate made by CMGP.

We evaluate two different utility functions: a linear utility $U_0(\gamma) = \gamma, U_1(\gamma) = \gamma - 4$ and a non-linear utility with $U_{0,i}(\gamma, x_i) = 1_{\gamma > E[Y(0)|X=x_i]}$ and $U_{1,i}(\gamma, x_i) = 1_{\gamma > (E[Y(0)_i|X=x_i]+4)}$, as the ATE for surface B is 4 (Hill, 2011). Table 1 shows AUC (linear) and AUC (threshold) corresponding to results under these two utilities. These results demonstrate the improved distribution estimates made by CCN compared to many techniques contribute to more accurate decisions, despite the fact that a homoskedastic Gaussian distribution is well matched to the specifications in BART and GAMLSS. CMGP and FCCN are very similar in their decision quality, despite CMGP's assumptions being well-matched to the data generation procedure. The evaluations of all these metrics can be regarded as a plug-in process for distribution estimators. We notice that PEHEs of these distribution methods may not be as competitive as some of the state-of-the-art CATE estimators

Table 2: Quantitative results on the EDU dataset (Section 5.3). FCCN achieves top marks on all metrics and outperforms CCN with statistical significance.

| Metrics | CCN | FCCN | GAMLSS | GANITE | BART | CMGP | CEVAE | CF |
|---|---|---|---|---|---|---|---|---|
| PEHE | $.392 \pm .049$ | $\mathbf{.296 \pm .042}$ | $.314 \pm .053$ | $1.253 \pm .181$ | $.534 \pm .042$ | $2.087 \pm .015$ | $1.911 \pm .351$ | $1.022 \pm .051$ |
| LL | $-2.178 \pm .024$ | $\mathbf{-2.125 \pm .022}$ | $-2.250 \pm .025$ | $-5.092 \pm .596$ | $-2.443 \pm .063$ | $-3.523 \pm .008$ | $-3.558 \pm .055$ | NA |
| AUC | $.933 \pm .026$ | $\mathbf{.953 \pm .014}$ | $.941 \pm .010$ | $.760 \pm .053$ | $.906 \pm .015$ | $.875 \pm .006$ | $.622 \pm .039$ | NA |

(Shalit et al., 2017; Hassanpour & Greiner, 2020a), as optimizing the mean is simpler and has an exact match with this metric. The strategy of estimating distributions can be flexibly adapted to various comparisons with trained models.

**Comparison to Policy Learning.** We next compare the proposed approaches to a policy learning approach, specifically policytree (Sverdrup et al., 2021), with full details in Appendix F. Policy learning is limited to pre-specified utility functions, so we set up two scenarios: one with a linear utility ($U_0(\gamma) = \gamma$, $U_1(\gamma) = \gamma - 4$), and one with a threshold (binary) utility ($U_0(\gamma) = 1_{\gamma > E[Y(0)]}$, $U_1(\gamma) = 1_{\gamma > E[Y(1)]}$). Since policytree only outputs its predicted optimal treatment, we compare on accuracy (predicted vs true optimal treatment). On the IHDP dataset, FCCN performs well on both linear and threshold utilities, achieving accuracies of 88.6% and 87.72% accuracy, respectively. However, policytree's accuracy drops from 76.9% to 57.6% when we switch from a linear to a threshold utility, signifying how much information is lost by binarizing the outcomes.

## 5.3 EDU

The EDU dataset is based on data collected from a randomized field experiment in India between 2011 and 2012 (Banerji et al., 2017; 2019). The experiment studies whether providing a mother with adult education benefits their children's learning. We define the binary treatment as whether a mother receives adult education and the continuous outcome as the difference between the final and the baseline test scores. After the preprocessing described in Appendix B, the sample size is 8,627 with 18 continuous covariates and 14 binary covariates, $x_i \in \mathbb{R}^{32}$.

We create a semi-synthetic case over the two potential outcomes according to the following procedures. We first train two neural networks, $f_{\hat{y}_0}(\cdot)$ and $f_{\hat{y}_1}(\cdot)$, on the observed outcomes for the control and treatment groups. The uncertainty model for the control and treatment group are based on a Gaussian distribution and an exponential distribution, respectively. The dependence of the outcome uncertainties on different distributions is intended to showcase the abilities of models that are able to adapt to different distribution families (e.g., CCN and FCN). We represent $m_i$ as an indicator of whether the mother has received any previous education, as we hypothesize the variability is higher for the mothers not educated previously[3]. Then, the potential outcomes are synthesized as,

$$Y(0)_i | x_i \sim f_{\hat{y}_0}(x_i) + (2 - m_i)N(0, 0.5^2); \quad Y(1)_i | x_i \sim f_{\hat{y}_1}(x_i) + (2 - m_i)\exp(2).$$

In this experiment, treatment group imbalance comes from two aspects of our data generation procedure. The first is from a treatment assignment model with propensity $\Pr(T_i = 1 | x_i) = [1 + \exp(-x_i^T \beta)]^{-1}$ where we assign large coefficients in $\beta$ to create imbalance. The other is from truncation, as we remove well-balanced subjects with estimated propensities in the range of $0.3 < \Pr(T_i = 1 | x_i) < 0.7$. We keep 1,000 samples for evaluation and use the rest for training. The full procedure is repeated 10 times for variability assessment. The utility function is customized for each subject to mimic personalized decisions. For subject $i$, $U_{0,i}(\gamma, x_i) = I(\gamma > v_i)$, and $U_{1,i}(\gamma, x_i) = I(\gamma > v_i + 1 - m_i)$ where $v_i \sim U(0, 1.5)$. The interpretation of this utility is that different mothers have different expectations of their children's improvements with threshold $v_i$. For the mothers without previous education, their expectations are higher by 1. This design coincides with the expectation that education should have a positive effect on outcomes in exchange for a finite cost, as we would only invest in the intervention if a positive return was expected.

---

[3]The variable $m$ indicating whether the mother has received previous education is included as a predictor in $X$; however, models are not aware how the $m$ variable relates to outcome variability and must learn this relationship. We simply assign it its own variable for ease of notation when describing how potential outcomes for the EDU semi-synthetic data are synthesized.

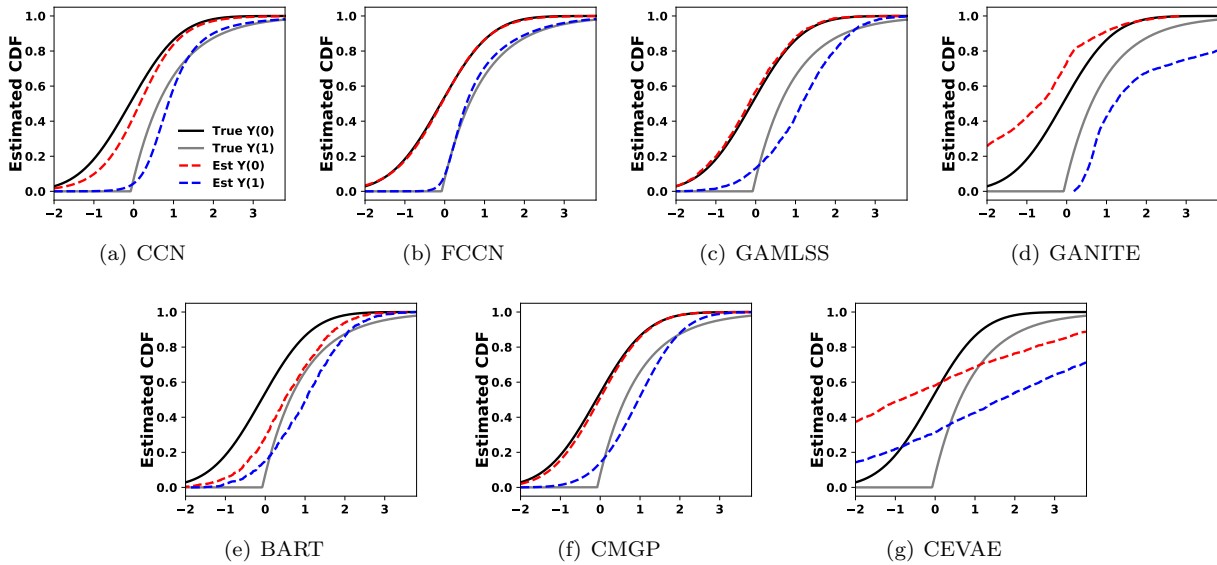

Figure 3: Visualization of estimated vs. true CDFs for a random EDU dataset sample (Section 5.3). FCCN estimates are the only ones that closely follow the true control and treatment group curves with high fidelity.

Results from this experiment are summarized in Table 2. CCN-based methods demonstrate their ability to flexibly model different distributions with FCCN providing improvement over CCN and top marks in all metrics. Figure 3 shows that FCCN is the only method able to recover both Gaussian *and* exponential distributions with high fidelity, which we believe contributes to its top performance. Making an optimal decision is highly dependent on how close a given method's estimated distribution aligns with the true values and all relevant heterogeneity. Thus, the AUCs follow their respective LLs with FCCN producing the best performance in both. Unlike the IHDP data, CMGP's modeling assumptions do not match the generative procedure, and its performance suffers. CEVAE makes two model misspecifications: (*i*) it assumes homogeneous Gaussian error, whereas the real outcomes arise from heteroskedastic Gaussian or exponential distributions, and (*ii*) it decodes the continuous covariates into a Gaussian distribution. Thus, CEVAE captures the marginal distributions well (visualized in Figure 1(h) in Appendix E) but does not provide helpful personalized suggestions.

Next, we randomly draw an EDU data sample and compare the estimated CDFs against the true CDFs in Figure 3 (see Figure S1 for additional samples). We find that CCN-based approaches are capable of faithfully recovering the true conditional CDFs, whereas the other methods have gaps in their estimation. GAMLSS is accurate on the control group but not the treatment group. This is partially due to `gamlss` package (Stasinopoulos et al., 2021) not supporting the exponential distribution with location shift, so skewed normal is chosen as the closest reasonable substitute. Overall, GAMLSS is flexible but requires precise specification on a case-by-case basis, whereas CCN can robustly use the same approach. In our experiments for GAMLSS, we must choose very close distributions and limit the uncertainty to the relevant variables or the package does not converge. Similarly, CMGP estimates the control CDF well as its Gaussian assumptions align with the underlying distribution of the control group. However, is not able to accurately capture the treatment CDF.

Lastly, we assess whether the heteroskedasticity of the outcomes is captured. The combination of $M = 0, 1$ and $T = 0, 1$ produces four uncertainty models. We visualize the predictive 90% interval widths in Figure 4. FCCN captures the bimodal nature of the interval widths. In contrast, GANITE only captures a small fraction of the difference between low and high variance cases. Both CEVAE (Louizos et al., 2017) and BART (Hill, 2011) fail to capture the heteroskedasticity and are not shown. For GAMLSS, we explicitly feed its uncertainty model with only $T$ and $M$ for it to converge effectively. It does not produce variability in interval widths. Overall, CCN and its variants produce higher quality ranges.

Table 3: The estimated LL under different simulated distributions (Section 5.4.2). [1] and [2] represent fitting GAMLSS with the true family and heteroskedastic Gaussian, respectively.

| | True Value | CCN | FCCN | BART | GAMLSS[1] | GAMLSS[2] |
|---|---|---|---|---|---|---|
| Gumbel | -2.87 | -3.67 ± .02 | **-3.56 ± .02** | -3.92 ± .06 | -3.67 ± .02 | -3.90 ± .05 |
| Gamma | -3.17 | -3.83 ± .04 | **-3.74 ± .06** | -3.97 ± .02 | -3.77 ± .02 | -3.95 ± .02 |
| Weibull | -2.87 | -3.41 ± .02 | **-3.32 ± .03** | -3.86 ± .12 | **-3.32 ± .04** | -3.71 ± .09 |

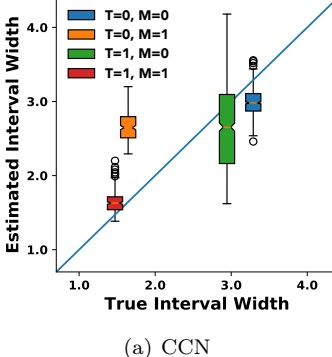
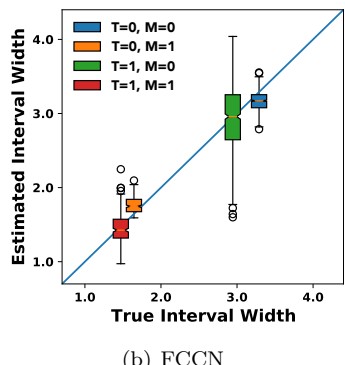
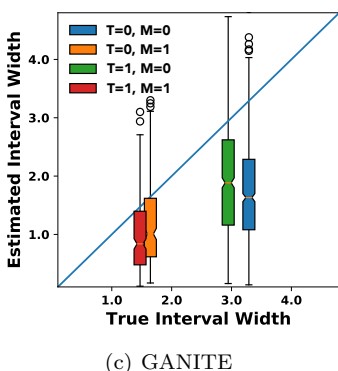

(a) CCN          (b) FCCN          (c) GANITE

Figure 4: Ability of methods to capture heteroskedastic outcomes of EDU data (Section 5.3) are demonstrated by plotting the estimated versus true 90% interval widths given four different combinations of $T$ and $M$. GANITE reflects the main trend of how uncertainties change with $T$ and $M$. FCCN can clearly discern the four scenarios by aligning its estimated interval widths and the true interval widths. Additional visualizations are provided in Appendix J.

## 5.4 Additional Comparisons and Properties

In this section, we include several additional experiments to reflect other properties of CCN and its adjustment to demonstrate their advantages in estimating the potential outcome distributions.

### 5.4.1 Estimating Multimodal Distributions

We choose to extend CN to estimate potential outcome distributions due to its ability to adapt to a variety of outcome families. We demonstrate this with a qualitative experiment that uses synthetically-generated multi-modal data distributions. Many of other methods, including CMGP, GAMLSS, BART, and CEVAE, cannot capture it with their existing structure, whereas CCN, FCCN, and methods like GANITE can. To succinctly summarize this experiment, only CCN and FCCN naturally adjust to the multi-modal space, as shown briefly in Figure 5. GANITE is aware of the mixtures but does weight them well. The other algorithms are not able to capture the mixture model. The data generating process is described in Appendix H.

### 5.4.2 Additional Distribution Tests

Next, we sketch out how different methods work on a variety of known outcome distributions, including Gumbel, Gamma, and Weibull distributions, with full details in Appendix G. The results in Table 3 are similar to the previously presented semi-synthetic cases, where CCN straightforwardly adapts to these distributions and FCCN provides additional improvements. In fact, FCCN even slightly outperforms GAMLSS even when GAMLSS is *provided the true outcome distribution*. GAMLSS does not come close to rivaling the performance of CCN when provided a flexible but not perfectly matched outcome distribution.

### 5.4.3 Sample Size and Convergence

Proposition 2 suggests CCN can asymptotically estimate the optimal value given a large sample size. We create a synthetic example with the logistic distribution to visualize it. The data generating process is

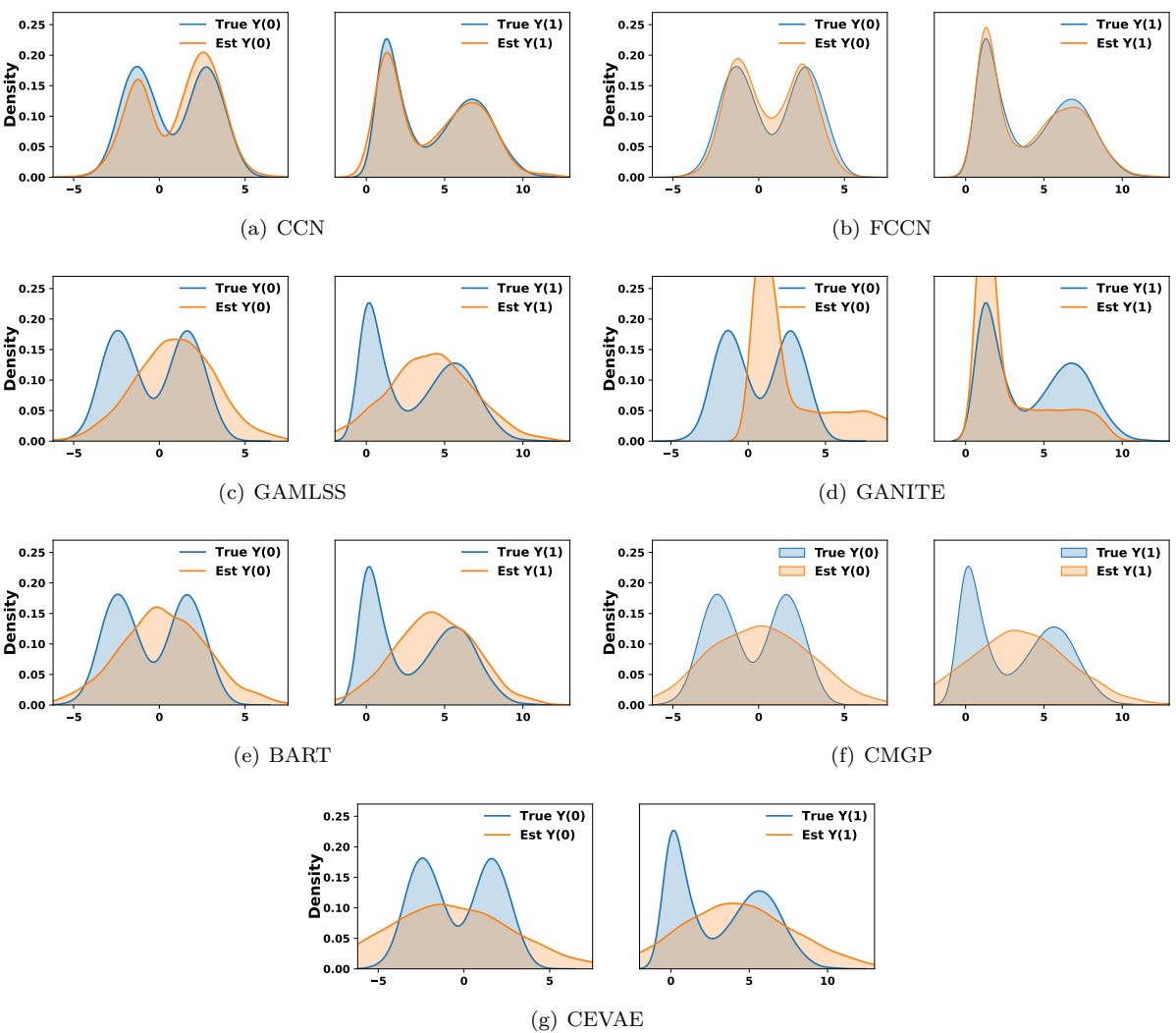

Figure 5: Visualization of each method's estimated density of the potential outcomes for synthetically generated multi-modal data (Section 5.4.1). Visually, CCN and FCCN outperform other methods in estimating the multimodal data. It appears GANITE learns that the data is non-unimodal in nature, but makes innacurate density estimates, greatly overestimating densities in some regions and underestimating density in others.

described in Appendix I. We vary the input data size and compare all methods on log-likelihood in Figure 6. We note that CCN and its variants all asymptotically approach the optimal value. All adjusted versions of CCN present faster convergence rates with FCCN dominating the curve. Gaussian methods (BART, CMGP[4], CEVAE) provide more stable approximations in smaller samples. Due to distributional mismatch, the optimal value can not be attained by those methods. Though GAMLSS has the correct family specification, it is restrained by the flexibility of the additive model to approach the optimal value. GANITE progresses slower and needs over 15,000 samples to generate competitive results.

### 5.4.4 Ablation Study

We include three components to account for treatment group imbalance. They are the Assign-loss (Assignment), the Wass-loss (Wass), and propensity stratification (PS). In this section, we inspect how CCN empirically benefits from each component. Additional details and visualizations can be found in Appendix J.

---

[4]CMGP convergence was only tested up to a certain sample size due to computational tractability of fitting the model to larger sample sizes ($n > 20,000$).

Table 4: Quantitative results on the IHDP dataset regarding different variants of CCN (Section 5.4.4).

| Metrics/Method | CCN | Wass | Assign | PS | Assign+PS | FCCN |
|---|---|---|---|---|---|---|
| PEHE | 1.59 ± .16 | 1.32 ± .17 | 1.42 ± .25 | 1.15 ± .10 | 1.22 ± .15 | **1.13 ± .14** |
| LL | -1.78 ± .02 | -1.65 ± .02 | -1.64 ± .03 | -1.67 ± .15 | -1.65 ± .02 | **-1.62 ± .02** |
| AUC (Linear) | .925 ± .011 | .938 ± .010 | .940 ± .010 | .918 ± .012 | .940 ± .010 | **.942 ± .010** |
| AUC (Threshold) | .913 ± .011 | .932 ± .011 | .932 ± .011 | .911 ± .012 | .934 ± .011 | **.935 ± .010** |

Table 5: Quantitative results on the EDU dataset regarding different variants of CCN (Section 5.4.4).

| Metrics/Method | CCN | Wass | Assign | PS | Assign+PS | FCCN |
|---|---|---|---|---|---|---|
| PEHE | .392 ± .049 | .324 ± .046 | .343 ± .041 | .400 ± .052 | .339 ± .039 | **.296 ± .042** |
| LL | -2.178 ± .024 | -2.128 ± .020 | -2.132 ± .023 | -2.171 ± .024 | -2.129 ± .029 | **-2.125 ± .022** |
| AUC | .933 ± .026 | .951 ± .013 | .946 ± .018 | .932 ± .022 | .952 ± .019 | **.953 ± .014** |

Table 4 and 5 summarize the results on the evaluating metrics in both the IHDP and EDU datasets. Overall, variants of CCN with adjustment more accurately estimate the potential outcomes. However, the aspects of how these components contribute vary according to their attributes. The propensity score stratification mainly facilitates information sharing between strata (Lunceford & Davidian, 2004); hence, it excels in the IHDP dataset where the imbalance is caused by removing a specific subset from the treatment group. However, on the conditional level the stratification can be regarded as a form of aggregation, which might hinder the precision. Therefore, we do not see much gain in distribution-based metrics or personalized decisions by solely including the propensity. The Assign-loss overcomes the confounding effect

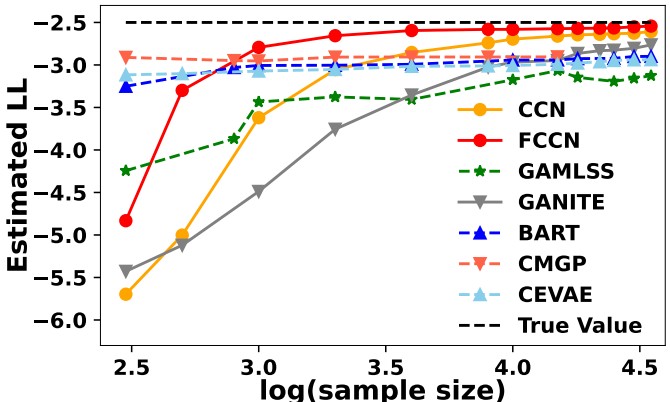

Figure 6: Convergence rate of models as a function of sample size (Section 5.4.3).

by extracting representation relevant to confounding effects, and we observe that it effectively increases the model performance in all metrics. The combination of Assign-loss and propensity stratification demonstrates the merits of these two approaches. The Wass-loss finds a representation that balances the treatment and control groups and improves both point and distribution estimates in the two datasets. Nevertheless, it does not account for the domain-specific information (Shi et al., 2019). FCCN achieves the best performance in all cases by a clear margin.

Moreover, we examine the proposed adjustment components in different scenarios in Figure 7. Figure 7(a) shows that CCN and its variants all asymptotically approach the optimal value with FCCN being the quickest. Figure 7(b) is made by varying the values of $\alpha$ or $\beta$. It suggests that parameter values that are either too large or too small are more likely to hurt models with only a single adjustment component, while FCCN is more robust to these changes. Figure 7(c) describes a case where irrelevant dimensions with standard Gaussian distributions are added to the covariate space. We observe that adding noise worsens the performance. In this case, the Assign-loss is more likely to overfit the propensity model with extra covariates containing noise only. Hence, adjustment through Wass-loss is preferable. In Figure 7(d), we vary the propensity model by changing its coefficient. Larger values represent less balanced space and larger confounding effects. In this setup, the Assign-loss is slightly better when the imbalance is more extreme, as a balanced representation becomes more challenging to obtain. Among them, FCCN is the most robust due to simultaneously considering the domain-invariant and specific information.

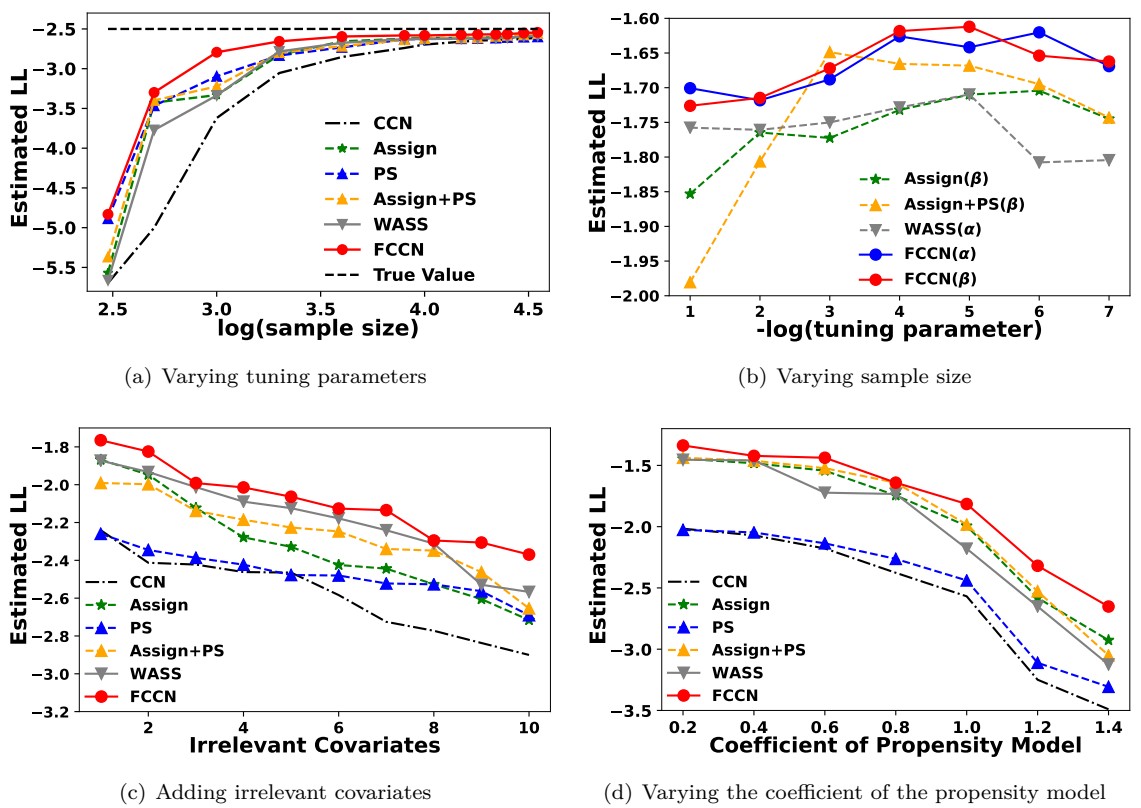

(a) Varying tuning parameters

(b) Varying sample size

(c) Adding irrelevant covariates

(d) Varying the coefficient of the propensity model

Figure 7: Estimated LL under different scenarios (Section 5.4.4). 7(a) demonstrates differences in convergence speed of CCN and its variants as a function of sample size. 7(b) depicts the performance of CCN variants given different tuning parameter values. 7(c) and 7(d) show that adding noise or treatment group imbalance hurts model performance. Under these different circumstances, FCCN remains the most robust.

## 6 Discussion

Here, we propose Collaborating Causal Networks (CCN): a novel framework to estimate conditional potential outcome *distributions*, backed by theoretical proofs and paired with an adjustment method that rectifies treatment group imbalance. Our experiments demonstrate that CCN is able to adapt to a variety of outcomes, including exponential family distributions and multi-modal distributions, thus empirically demonstrating that CCN is effective in inferring *full potential outcomes*. Additionally, incorporation of our proposed adjustment technique in FCCN is relatively robust with regards to treatment group imbalance in synthetic and semi-synthetic experiments. We note that improving distribution estimates leads to improved decision-making even without *a priori* access to utility functions by comparing to policy learning. In all of our evaluations, FCCN meets or exceeds the current state-of-the-art methodology for potential outcomes distribution estimation, and asymptotically approaches our theoretical claims.

CCN and its variants are competitive with but do not outperform methods that make modeling assumptions that are perfectly aligned with the true data generating distribution, as is the case with CMGP and the IHDP dataset. Therefore, if the true data generating process is known prior to modeling then other methods may be preferred over CCN. However, we note that access to the true data generating mechanism is not common in practice and that these processes may be quite complex, in which case we would still expect CCN and its variants to asymptotically capture outcome distributions regardless of form. The Lipschitz continuous assumption is a potential limitation, albeit a small one. A Lipschitz CDF entails PDF without point masses, which prohibits some distributions, such as zero-inflated probability distributions, but is mild in practice under an additive noise model. Like many causal inference models, the proposed approach is dependent on key assumptions. In this case, the primary assumptions are summarized by the strong ignorability

assumptions. Finally, as with any method with relevance to healthcare or precision medicine, we emphasize this framework should not be used alone to guide clinical practice, suggest treatments, or recommend any health-related actions without consultation and close collaboration with medical professionals. When applying CCN, a practitioner should take care to evaluate the reasonableness of the assumptions, especially that of unconfoundedness, by consulting with domain experts about the features included as predictors in the model.

## Acknowledgments

We acknowledge the anonymous reviewers for their helpful suggestions and comments.

Research reported in this manuscript was supported by the National Institute of Biomedical Imaging and Bioengineering and the National Institute of Mental Health through the National Institutes of Health BRAIN Initiative under Award Number R01EB026937. The contents of this manuscript are solely the responsibility of the authors and do not necessarily represent the official views of any of the funding agencies or sponsors.

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

# A    Proofs of Propositions 1 and 2

Zhou et al. (2021) assume that the covariate distributions are the same for training and generalization for CN. In observational studies, the training spaces $p(x|T = 1)$ and $p(x|T = 0)$ differ from the evaluation space $p(x)$. Thus, the central challenge of migrating the properties of CN to CCN is demonstrating robustness of CCN with respects to this covariate space mismatch.

First, we explore the properties of CN and CCN in the presence of covariate space mismatch. Second, under assumptions commonly made in the causal inference literature, we expand on how CCN can overcome covariate space mismatch. For completeness, we restate the two propositions from the main section of the paper. Note that both claims are made with respects to the *full* covariate space, $\forall x$ such that $p(x) > 0$.

**Proposition S1** (Optimal solution for $g_0$ and $g_1$). *When the support of the outcomes is a subset of the support of $z$, or as $z$ covers the whole outcome space, the functions $g_0$ and $g_1$ that minimize g-loss* are optimal when they are equivalent to the conditional CDF of $Y(0)|X = x$ and $Y(1)|X = x$, $\forall x$ such that $p(x) > 0$.*

**Proposition S2** (Consistency of $g_0$ and $g_1$). *Assume the ground truth CDF functions for $T \in \{0, 1\}$ are Lipschitz continuous with respect to both the features $X$ and the potential outcomes $\{Y(0), Y(1)\}$ and the support of the outcomes is a subset of the support of $z$. Denote the ground truth functions as $g_0^*$ and $g_1^*$. As $n \to \infty$, the finite sample estimators $g_0^n$ and $g_1^n$ have the following consistency property: $d(g_0^n, g_0^*) \to_P 0; d(g_1^n, g_1^*) \to_P 0$ under some metrics $d$, such as the $\mathbb{L}_1$ norm, and with the space searching tool $z$ being able to cover the full outcome space.*

## A.1    Claims from Zhou et al. (2021) with and without Space Mismatch

We now restate two CN propositions from Zhou et al. (2021) under the non-causal setting where there is no space mismatch between the training and the evaluation spaces.

**Proposition S3** (Optimal solution for $g$ from Zhou et al. (2021)). *Assume that $f(q, x)$ approximates the conditional $q^{th}$ quantile of $Y|X = x$ (inverse CDF, not necessarily perfect). If $f(q, x)$ spans $\mathbb{R}^1$, then a $g$ minimizing equation 1 is optimal when it is equivalent to the conditional CDF, or $\Pr(Y < y|X = x) = g(y, x)$, $\forall y \in \mathbb{R}$, $\forall x$ such that $p(x) > 0$.*

**Proposition S4** (Consistency of $g$ from Zhou et al. (2021)). *Assume the true CDF function $g^*$ satisfies Lipschitz continuity with respect to $x$ and outcome $y$. As $n \to \infty$, the finite sample estimator $g^n$ has the following consistency property: $d(g^n, g^*) \to_P 0$ under some metric $d$ such as $\mathbb{L}_1$ norm and with $f$ capable of searching the full outcome space.*

Proposition S3 demonstrates that a fixed point solution estimates the correct distributions. Proposition S4 states that the optimal learned function asymptotically estimates the true distributions. However, they do not answer whether these properties hold on a new space that differs from the original training space. We address this existing limitation by developing Proposition S5 which shows that these properties can still be retained given a certain type of space mismatch. For generality, we define the training space as $p(x)$ and the new space as $p'(x)$.

**Proposition S5** (The dependency of CN on $p(x)$). *If the conditional outcome distribution $Y|X = x$ remains invariant between $X \sim p(x)$ and $X \sim p'(x)$ (covariate shift), and $p(x) > 0 \implies p'(x) > 0$, the solutions in Propositions S3 and the consistency in Proposition S4 also generalize to the new space where $p'(x) > 0$.*

*Proof of Proposition S5.* With the Propositions S3 and S4, we observe that $g$ estimates the conditional distribution of $Y|X = x$ in the space where $p(x) > 0$.

Next, we generalize it to a new space $p'(x)$. Given the condition $p(x) > 0 \implies p'(x) > 0$, for any $x$ in evaluation space with $p'(x) > 0$, it is covered in the training space where $p(x) > 0$. From Proposition S3 and S4, we know that for such $x$, the optimum can be obtained. The covariate shift assumption on the invariance of outcome distributions then guarantees that the optimum of such $x$ in the training space is also the optimum in the new space.

Therefore, each point $x$ in the new space with $p'(x) > 0$ can obtain their optimum through training the model on the training space $p(x)$. □

## A.2 Overcoming Covariate Space Mismatch for CCN

Proposition S5 enables us to extend the optimum of CN to new spaces given two conditions: the *covariate shift* and the *space overlap* $p(x) > 0 \implies p'(x) > 0$. Our main task is to show how they hold in causal settings. First, we give a weaker version of Propositions 1 and 2 as a direct result from Proposition S3 and S4 without accounting for the mismatch between the training and generalization spaces.

**Claim S1** (Potential outcome distributions on each treatment space)**.** *Given all the conditions in Proposition S3 and S4, an optimal solution for $g_0$ and $g_1$ in g-loss* in equation 4 are the true CDFs of $Y(0)|X = x$, $T = 0$, $\forall x$, such that $p(x|T = 0) > 0$; and $Y(1)|X = x$, $T = 1$, $\forall x$, such that $p(x|T = 1) > 0$. The finite sample estimators $g_0^n$ and $g_1^n$ can also consistently estimate these CDFs.*

*Discussion on Claim S1.* We only discuss the first part of Claim S1 for $T = 0$ without loss of generality, while optimizing $g_0$ involves updating parameters using batches with $t = 0$. The other group can be shown using identical steps.

A direct conclusion from Propositions S3 and S4 is that the optimal $g_0$ is the fixed point solution, and finite sample estimator $g_0^n$ consistently estimates the true CDF of $Y|X = x$, $T = 0$, $\forall x$, such that $p(x|T = 0) > 0$. This is claimed for the full conditional distribution $Y|X = x$, $T = 0$ rather than for the potential outcome $Y(0)$. By virtue of the treatment consistency (Assumption 2), the following two outcomes are identically distributed: $Y|X = x$, $T = 0 \iff Y(0)|X = x$, $T = 0$. Therefore, we can successfully establish the estimators for potential outcome distributions, but currently limited to each treatment subspace. □

As mentioned above, we need two conditions to generalize Claim S1 back to the full space with density $p(x)$. The covariate shift has been explicitly described in (Johansson et al., 2020). Covariate shift is a straightforward consequence of Assumption 3, and we state it for clarity.

**Lemma S1** (Covariate shift)**.** *The conditional distributions of potential outcomes $Y(0)|X = x$ and $Y(1)|X = x$ are independent of the covariate space in which they are located.*

*Discussion on Covariate Shift.* The ignorability states that $[Y(0), Y(1)] \perp T|X$. It implies that $\forall y(0), y(1), x$, the three following density functions are equivalent: $p(y(0), y(1)|X = x, T = 0) = p(y(0), y(1)|X = x, T = 1) = p(y(0), y(1)|X = x)$.

This supports that the potential outcome distributions are invariant to the treatment groups. □

We next show the condition of the space overlap, which is a straightforward consequence of Assumption 1.

**Lemma S2** (Positivity relating to the space overlap)**.** *Under Assumption 1, the equivalent condition holds: $p(x) > 0 \iff p(x|T = 0) > 0$, and $p(x) > 0 \iff p(x|T = 1) > 0$.*

*Proof of Lemma S2.* The positivity in Assumption 1 claims that $\forall x$, $0 < \Pr(T = 1|x) < 1$. Then for each $x$ where $p(x) > 0$, we can find a constant $1 > C_x > 0$ that satisfies $\Pr(T = 1|x) > C_x$.

By Bayes rule, $p(x|T = 1) = \Pr(T = 1|x)p(x)/\Pr(T = 1) > C_x p(x) > 0$. Then $p(x) > 0 \implies p(x|T = 1) > 0$. From the other direction, if $p(x|T = 1) = \Pr(T = 1|x)p(x)/\Pr(T = 1) > 0$, then each component on the right hand side needs to be positive. Therefore, $p(x|T = 1) > 0 \implies p(x) > 0$. We can use the same procedure to verify the condition for $T = 0$. □

With Lemma S1 and S2 satisfying the conditions in Proposition S5, we acquire that the space mismatch given the standard causal assumptions is the specific type that does not impact the large sample properties of CCN. As a result, Propositions 1 and 2 naturally follow from these two Lemmas (or alternatively, following from Assumption 1 and 3). Thus, under the standard assumptions in causal inference, CCN will capture the full potential outcome distributions. This procedure is not limited to a binary treatment condition and is extendable to multiple treatment setups.

## B   Semi-Synthetic Data Generation

### B.1   IHDP

Simulated replicates for the IHDP data were downloaded directly from `https://github.com/clinicalml/cfrnet`, which were used to generate the results of WASS-CFR (Shalit et al., 2017) and CEVAE (Louizos et al., 2017). The IHDP dataset does not contain personally identifiable information or offensive content.

### B.2   Education Data

The raw education data corresponding to the EDU dataset were downloaded from the Harvard Dataverse[5], which consist of 33,167 observations and 378 variables. The dataset does not contain personally identifiable information or offensive content.

We pre-process the data by combining repetitive information and deleting covariates with over 2,500 missing values. This results in a processed dataset containing 8,627 observations. The function $f_{\hat{y_1}}(\cdot)$ and $f_{\hat{y_0}}(\cdot)$ are learned from the observed outcomes for the treatment and control groups. They are both designed as single-hidden-layer neural networks with 32 units and sigmoid activation functions (Han & Moraga, 1995). A logistic regression model with coefficient $\beta = [\beta_1, ..., \beta_{28}]$ and propensity score $\Pr(T = 1 | X = x_i) = \frac{1}{1+exp(-x_i'\beta)}$ are used to generate treatment labels and mimic observational study data. The coefficients are randomly generated as $\beta_i \sim U(-0.8, 0.8)$.

## C   Detailed Method Implementations

The models CCN, CEVAE and GANITE were trained and evaluated on the various datasets described in this work on a machine with a single NVIDIA P100 GPU. The R-based methods BART, CF, and GAMLSS were trained and evaluated on a machine with an Intel(R) Xeon(R) Gold 6154 CPU. The CMGP model was trained and evaluated on a machine with an Intel Core i7 10$^\text{th}$ generation processor and a NVIDIA GeForce RTX 3090 GPU.

**CCN and FCCN**

The implementation of CCN and all its variants are based on the code base for CN (Zhou et al., 2021), which is provided at `https://github.com/thuizhou/Collaborating-Networks` under the MIT license. Functions $g_0$ and $g_1$ follow the structures of $g$ in Zhou et al. (2021). We implement the full collaborating structure and find the optimization to be harder when added with regularization terms. Therefore, we fix $f$ as a uniform distribution covering the range of the observed outcomes. This is referred to as the "g-only" setup in Zhou et al. (2021), and is easier to optimize with only a marginal loss in accuracy.

In FCCN, we introduce two latent representation $\phi_A(\cdot)$ and $\phi_W(\cdot)$. We set their dimensions to 25. They are both parameterized through a neural network with a single hidden layer of 100 units. The Wasserstein distance in Wass-loss is learned through $D(\cdot)$ which is a network with two hidden layers of 100 and 60 units per layer. We adopt the weight clipping strategy with threshold: (-0.01, 0.01) to maintain a Lipschitz constraint (Arjovsky et al., 2017). The hyperparameters $\alpha$ and $\beta$ are tuned for FCCN by selecting candidate values that do not significantly impact the log-likelihood calculated upon the observed outcomes, with the idea being to balance fitting the observed data while also encouraging generalization through non-zero $\alpha$ and $\beta$ values. We propose a few candidate values for $\alpha$ and $\beta$ as: 5e-3, 1e-3, 5e-4, 1e-4, 5e-5, 1e-5, as we do not want these values to be too large to overpower the part of the objective that learns the empirical distribution. Then, we perform a grid search to determine the hyper-parameters. Based on the results on the first few simulations, we fix $\alpha = $ 5e-4 and $\beta = $ 1e-5 in IHDP experiments and $\alpha = $ 1e-5 and $\beta = $ 5e-3 in EDU experiments. We find this specification to consistently improve the performance over regular CCN. To assess the potential outcome distributions, and take expectation over a defined utility function and draw 3,000 samples for each test data point with the learned $g_0$ and $g_1$. *The code for CCN and its adjustment will be public on GitHub under the MIT license if the manuscript is accepted.*

---

[5]`https://dataverse.harvard.edu/dataset.xhtml?persistentId=doi:10.7910/DVN/19PPE7`

**BART (Chipman et al., 2010)** The implementation of BART uses the `R` package `BayesTree` (Chipman & McCulloch, 2016) under the GPL (>= 2) License. We use the default setups in its model structure. Chipman et al. (2007) suggest that BART's performance with a default prior is already highly competitive and is not highly dependent on fine tuning. We set the burn-in iteration to 1,000. We draw 1,000 random samples per individual to access their posterior predicted distributions, as the package stores all the chain information and is not scalable for large data.

**CEVAE (Louizos et al., 2017)** CEVAE is implemented with the publicly available code from `https://github.com/rik-helwegen/CEVAE_pytorch/`. We follow its default structure in defining encoders and decoders. The latent confounder size is 20. The optimizer is based on ADAM with weight decay according to Louizos et al. (2017). We use their recommended learning rate and decay rate in IHDP experiments. In EDU experiments, the learning rate is set to 1e-4, and the decay rate to 1e-3 after tuning. We draw 3,000 posterior samples to access the posterior distributions.

**GANITE (Yoon et al., 2018)** The implementation of GANITE is based on `https://github.com/jsyoon0823/GANITE` under the MIT License. The model consists of two GANs: one for imputing missing outcomes (counterfactual block) and one for generating the potential outcome distributions (ITE block). Within each block, they have a supervised loss on the observed outcomes to augment the mean estimation of the potential outcomes. We use the recommended specifications in Yoon et al. (2018) to train the IHDP data. In the EDU dataset, the hyperparameters for the supervised loss are set to $\alpha = 2$ (counterfactual block) and $\beta = 1e\text{-}3$ (ITE block) after tuning.

**CF (Wager & Athey, 2018)** The implementation of CF uses the `R` package `grf` (Tibshirani et al., 2020) under the GPL-3 License. We specify the argument `tune.parameters="all"` so that all the hyperparameters are automatically tuned.

**GAMLSS (Hohberg et al., 2020)** The implementation of GAMLSS uses the `R` package `gamlss` (Rigby & Stasinopoulos, 2005) under the GPL-3 License. Since this method uses likelihood to estimate its parameters and often does not converge under complex models, we feed the location, scale, and shape models with relevant variables only. In location models, we fit all continuous variables with penalized splines. In scale and shape models, we use relevant variables in their linear forms. This choice is based on consideration of the balance between representation power and model stability.

**CMGP (Alaa & van der Schaar, 2017)** The implementation of CMGP uses the Python package `cmgp` under the 3-Clause BSD License. For all experiments in which CMGP was used, we use a maximum iteration count (`max_gp_iterations` parameter) of 100 when fitting the CMGP model.

## D    Enforcing a Monotonicity Constraint

The learned CCN model should have a monotonic property that $g(x, z + \epsilon) \geq g(x, z)$, $\forall \epsilon \geq 0$. In our experiments, this condition is learned with a neural network architecture with our training scheme. We do not see any non-trivial violations of this requirement. If required, though, this scheme can be enforced by modifying the neural network structure. One way of accomplishing this goal is to use a neural network with the form,

$$g(x, z) = \sum_{j=1}^{J} \text{softmax}(g_x^w(x))_j \sigma(g_x^b(x)_j + \exp(g_x^a(x)_j)z).$$

Here, $\sigma(\cdot)$ represents the sigmoid function. $g_x^w$, $g_x^b$, and $g_x^a$ are all neural networks that map from the input space to a $J$-dimensional vector, $\mathbb{R}^p \to \mathbb{R}^J$. In this case, the formulation of the outcome is still highly flexible but becomes a mixture of sigmoid functions. As the multiplier on $z$ is required to be positive, each individual sigmoid function is monotonically increasing as a function of $z$. Because the weight on each sigmoid is positive, this creates a full monotonic function as a function of $z$.

We implement this structure and find that it is competitive with a more standard architecture but is more difficult to optimize. As it is not empirically necessary to implement this strategy, we prefer the standard architecture in our implementations.

# E    Additional CDF Visualizations

We provide additional visualizations for the estimated potential outcome distributions with each method in Figure S1 based on another data point, which agrees with the results visualized in Figure 3. The two variants of CCN are capable of capturing the main shape of the true CDF curves, including the asymmetry of the exponential distribution, with higher fidelity. GAMLSS is less accurate in the treatment group due to using skewed normal for the exponential distribution with location shifts. GANITE's two-GAN structures are highly reliant on data richness for accurate predictions (Yoon et al., 2018), so it falls short in cases with greater treatment group imbalance. BART provides reasonable estimates but struggles with the misspecification by its Gaussian form. Similar to GAMLSS, CMGP captures the control group CDF well but does not approximate the treatment curve with high fidelity. CEVAE captures the overall marginal distribution for the potential outcomes as shown in Figure 1(h), but fails to discern the heterogeneity in conditional distributions in figure S1(g).

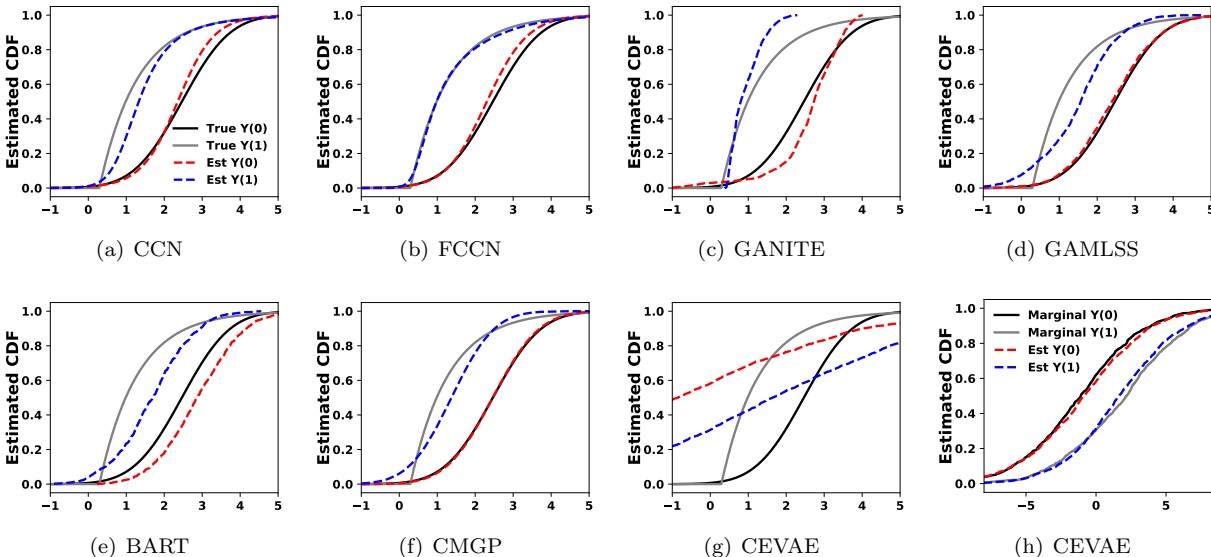

Figure S1: Additional visualizations of estimated vs. true CDFs for a random EDU dataset sample (Section 5.3, Section E). The two variants of CCN give good distribution estimates, while other methods provide less accurate estimates. By comparing the posterior distributions of CEVAE against the conditional distribution and marginal distribution of the ground truth in Figure S1(g) and Figure S1(h), we conclude that CEVAE primarily captures the marginal distribution in this study.

# F    Policy Learning

In decision making, the core difference between a traditional policy learning method and a distribution learning method is whether the utility is determined in advance. Though a policy learning method can tailor decisions based on different utilities, it is at the cost of fitting a new model to each proposed utility. Regardless of the inconvenience in computation, we discuss below another shortcoming of traditional policy learning methods. To train a traditional policy learning approach, the first step is often to convert the raw outcome to the observed utility. While this is less problematic for bijective transformations, it might result in information loss if we deal with discretized utilities.

To demonstrate, we compare FCCN to policytree with its published package (Athey & Wager, 2021; Sverdrup et al., 2021) on IHDP. We propose two types of utilities. They include a linear utility, $U_0(\gamma) = \gamma, U_1(\gamma) = \gamma - 4$, and a threshold utility, $U_0(\gamma) = 1_{\gamma > E[Y(0)]}, U_1(\gamma) = 1_{\gamma > E[Y(1)]}$. Since the `policytree` package only outputs the decision, we use accuracy as the evaluation metric. The results are summarized in Table S1. FCCN consistently makes more correct decisions. The information loss in the threshold utility negatively impacts

Table S1: Comparing the accuracy of policy learning between FCCN and policytree (Section F).

| Utility/Method | FCCN % | policytree % |
|---|---|---|
| Linear | $88.60 \pm 1.09$ | $76.86 \pm 1.03$ |
| Threshold | $87.72 \pm 1.19$ | $57.64 \pm .71$ |

policytree. We note that a threshold utility drastically reduces the available information by converting a continuous scale to a binary scale.

Fundamentally, these two methods are different and they address similar problems from different perspectives. There might be some possibilities that we could combine their merits. Hence, we will consider exploring their interactions more in future work.

## G   Additional Distribution Tests

To further illustrate the potential of CCN to model different types of distributions with high fidelity, we simulate potential outcomes from three extra distributions to assess its adaptability. For comparison we include GAMLSS, BART, CCN, and FCCN. The assessment is based on log likelihood (LL) to reflect the closeness to the true distributions. We simulate the covariate spaces and treatment labels with the following procedures:

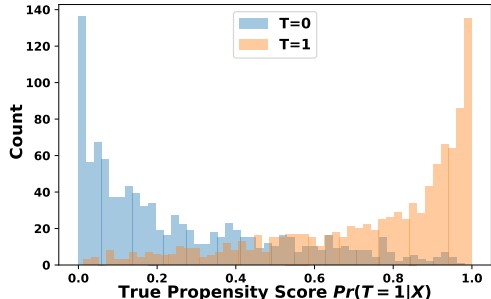

Figure S2: The limited propensity score overlap between two treatment groups (Section G). This indicates a severe treatment group imbalance.

Covariates:

$$x_i = (x_{1,i}, \cdots, x_{10,i})^\intercal, \ x_{j,i} \overset{i.i.d.}{\sim} \ N(0,1);$$

Treatment assignment:

$$\Pr(T_i = 1|x_i) = \frac{1}{1 + \exp(-x_i^\intercal \beta)}, \beta = (0.8, ..., 0.8)^\intercal$$

Given the magnitude of $\beta$, we have created a covariate space with limited overlap between two treatment groups as shown in Figure S2. With limited sample size, it also helps us evaluate the robustness of our method when positivity in Assumption 1 is possibly violated. Then we specify three scenarios with sufficient nonlinearity added to the potential outcome generating processes. We generate 2,000 data points in each case, and the variability assessment is based on 5-fold cross validation.

**Gumbel Distribution:**

$$Y(0)_i|x_i \sim \text{Gumbel}\left(5\left[\sin(\sum_{j=1}^{10} x_{j,i})\right]^2, 5\left[\cos(\sum_{j=1}^{10} x_{j,i})\right]^2\right); \ Y(1)_i|x_i \sim \text{Gumbel}\left(5\left[\cos(\sum_{j=1}^{10} x_{j,i})\right]^2, 5\left[\sin(\sum_{j=1}^{10} x_{j,i})\right]^2\right).$$

**Gamma Distribution:**

$$Y(0)_i|x_i \sim \text{Gamma}\left(4\sqrt{\left|\sin(\sum_{j=1}^{5} x_{j,i}) + \cos(\sum_{j=6}^{10} x_{j,i})\right|} + 0.5, \ 2\sqrt{\left|\cos(\sum_{j=1}^{5} x_{j,i}) + \sin(\sum_{j=6}^{10} x_{j,i})\right|}\right);$$

$$Y(1)_i|x_i \sim \text{Gamma}\left(4\sqrt{\left|\cos(\sum_{j=1}^{5} x_{j,i}) + \sin(\sum_{j=6}^{10} x_{j,i})\right|} + 0.5, \ 2\sqrt{\left|\sin(\sum_{j=1}^{5} x_{j,i}) + \cos(\sum_{j=6}^{10} x_{j,i})\right|}\right).$$

**Weibull Distribution:**

$$Y(0)_i|x_i \sim \text{Weibull}\left(5\sqrt{\left|\sin(\sum_{j=1}^{5} x_{j,i}) + \cos(\sum_{j=6}^{10} x_{j,i})\right|}, \ 2\sqrt{\left|\cos(\sum_{j=1}^{5} x_{j,i}) + \sin(\sum_{j=6}^{10} x_{j,i})\right|} + 0.2\right);$$

$$Y(1)_i | x_i \sim \text{Weibull}\left(5\sqrt{\left|\left|\cos(\sum_{j=1}^{5} x_{j,i}) + \sin(\sum_{j=6}^{10} x_{j,i})\right|\right|},\ 2\sqrt{\left|\left|\sin(\sum_{j=1}^{5} x_{j,i}) + \cos(\sum_{j=6}^{10} x_{j,i})\right|\right| + 0.2}\right).$$

## H    Estimating Multimodal Distributions

We use the following steps to generate the data from a multimodal distribution:

Covariates: $x_i \overset{i.i.d.}{\sim} N(0,1)$;

Treatment assignment: $\Pr(T_i = 1) = 1_{x_i > 0}$;

Mixture parameter: $\phi_i \sim i.i.d,\ \text{Bernoulli}(0.5)$;

Potential outcomes: $Y(0)_i | x_i \sim \phi_i N(-2,1) + (1-\phi_i)N(2,1) + x_i,\ Y(1)_i | x_i \sim \phi_i N(6, 1.5^2) + (1-\phi_i)\exp(1) + x_i$.

The control group is a mixture of two Gaussian distributions, and the treatment group is a mixture of Gaussian and exponential distributions. The mixture information is not given to any model, and we simply use their original form to approximate the distributions. Each model is trained using 1,600 simulated samples.

## I    Sample Size and Convergence

We create an example with the logistic distribution to visualize the performance of models with respect to sample size. We simulate 40,000 samples in total and hold out 2,000 for evaluation based on log likelihood (LL) with the following procedures:

Covariates: $x_i = (x_{1,i}, x_{2,i}, x_{3,i})^\intercal,\ x_{j,i} \overset{i.i.d.}{\sim} N(0,1)$;

Treatment assignment: $\Pr(T_i = 1 | x_i) = \frac{1}{1 + exp(-x_i^\intercal \beta)},\ \beta = (2, 2, 2)^\intercal$;

Scale parameter: $\sigma_i = |x_{1,i} + x_{2,i} + x_{3,i}| + 0.5$;

Location parameter: $\mu(0)_i = \sin(x_{1,i}\pi + x_{2,i}\pi) + \sin(x_{3,i}\pi),\ \mu(1)_i = \cos(x_{1,i}\pi + x_{2,i}\pi) + \cos(x_{3,i}\pi)$;

Potential outcomes: $Y(0)_i | x_i \sim \text{Logistic}(\mu(0)_i, \sigma_i), Y(1)_i | x_i \sim \text{Logistic}(\mu(1)_i, \sigma_i)$.

## J    Ablation Study

First, we provide an additional visualization in Figure S3 to reflect how well each adjustment improves the the uncertainty estimates of EDU data. FCCN not only captures the heteroskedasticity, but also reduces uncertainty by exhibiting narrower uncertainty bars in the corresponding plots.

### J.1    Additional Imbalance Adjustment Study

We give another motivating example to visualize the added robustness provided by our adjustment scheme. The visualizations in Figure 7(b), 7(c) and 7(d) are all based on the same synthetic data, described below. We use the same covariate space and treatment assignment mechanism in Appendix G. However, we posit a nonstandard distribution with its location model as a trigonometric function, and outcome uncertainty model as a heteroskedastic Beta distribution. We generate 2,000 data points in total, with 8/2 split for training and testing. The detailed synthetic procedure for the outcomes is described below:

$$Y(0)_i | x_i \sim \text{Beta}\left(\frac{\sum_{j=1}^{5}|x_{j,1}|}{5}, \frac{\sum_{j=6}^{10}|x_{j,1}|}{5}\right) + \sin\left(\sum_{j=1}^{10} x_{j,i}\right);\ Y(1)_i | x_i \sim \text{Beta}\left(\frac{\sum_{j=6}^{10}|x_{j,1}|}{5}, \frac{\sum_{j=1}^{5}|x_{j,1}|}{5}\right) + \cos\left(\sum_{j=1}^{10} x_{j,i}\right).$$

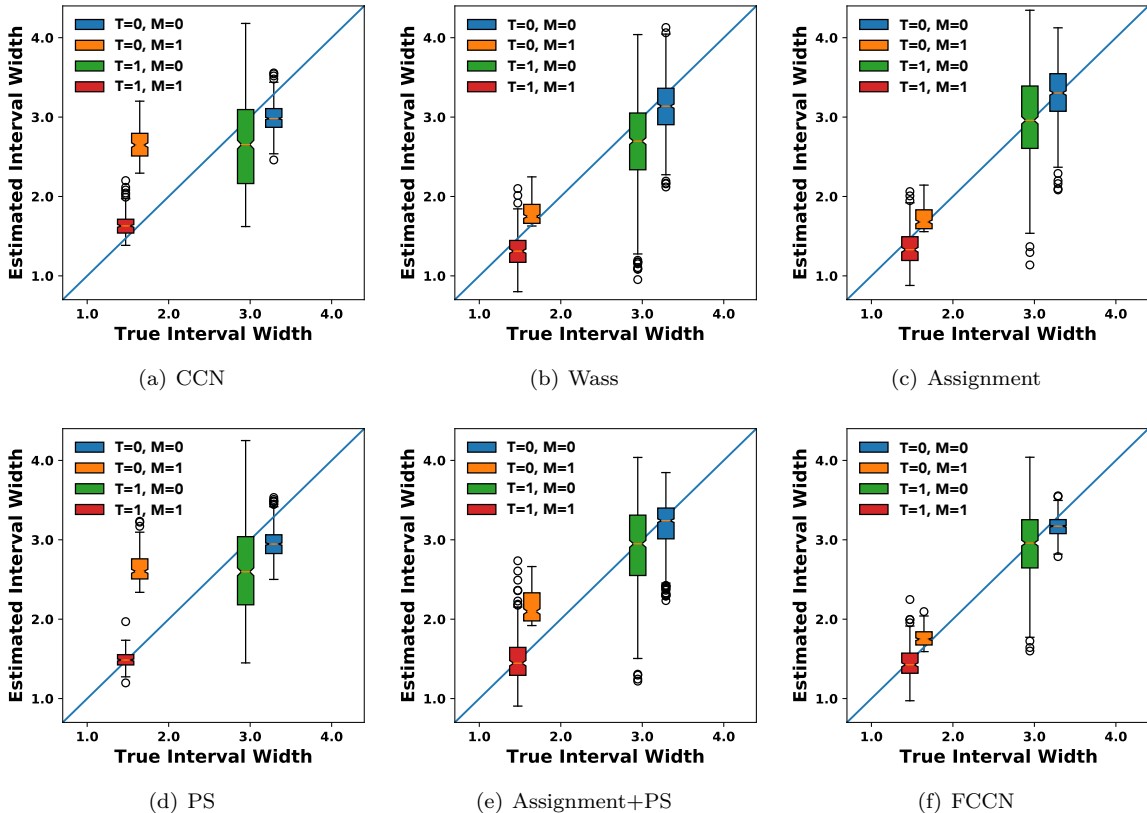

(a) CCN      (b) Wass      (c) Assignment

(d) PS      (e) Assignment+PS      (f) FCCN

Figure S3: The ability of different variations of CCN to capture heteroskedastic outcomes of EDU data (Section 5.3) are demonstrated by plotting the estimated versus true 90% interval widths given four different combinations of $T$ and $M$ in an additional ablation study (Section J). FCCN not only captures the heteroskedasticity, but also reduces the uncertainty by exhibiting narrower regions of uncertainty.

We visualize the performance of different adjustment schemes in scatter plots where the $x$ axis corresponds the true propensity score of each point. Figure S4 depicts the absolute difference between the inferred CATEs and true CATEs. Lower vertical positions represent lower errors. We observe that the performance deteriorates in each method if a point is close to either of the two boundaries (extreme propensity scores), which are areas in which models generally struggle the most in observational studies. Compared with the baseline CCN, each adjustment scheme by itself lowers the error to some extent. Among them, Wass-CCN, Assign-CCN and FCCN are able to reduce the average error by over 50 %. The propensity stratification (PS) adjustment can effectively reduce bias when there is more homogeneity within each stratum. However, severe imbalance in this case only gives homogeneity in the strata where propensity is around 0.5. Hence, PS provides limited benefits. In contrast, Wass-loss and Assign-loss seek new representations to either rectify group level imbalance or exclude confounding effects. They prove to be more effective in the regions of imbalance, which represent the majority of the data points.

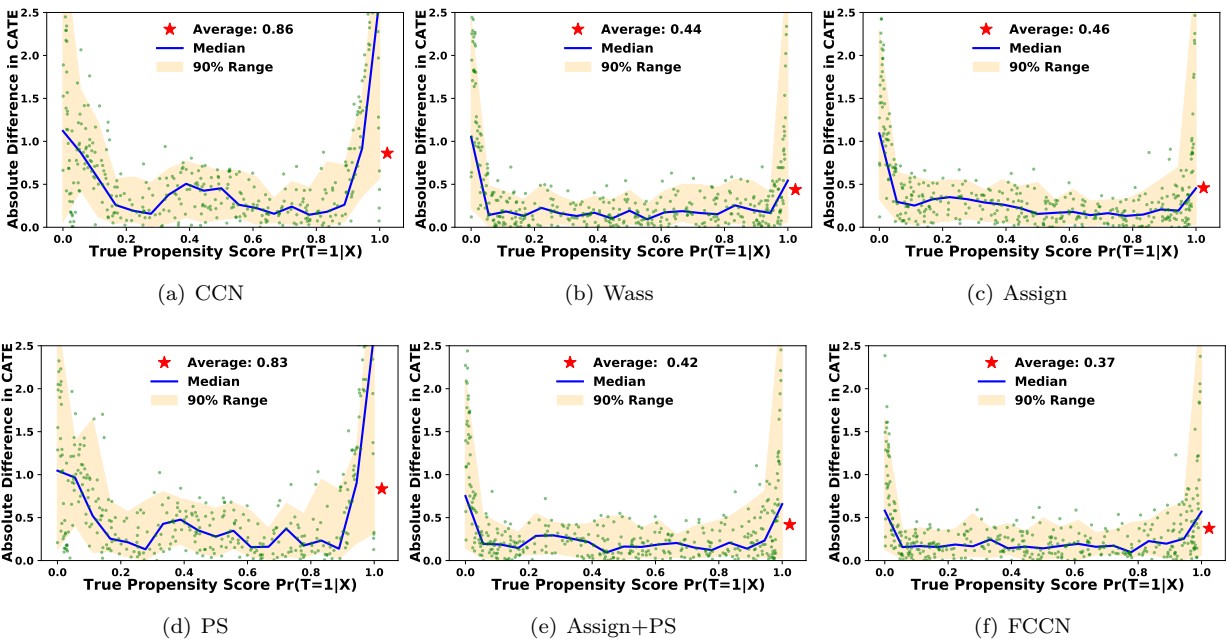

Figure S4: Scatter plot of propensity scores (x-axis) versus the absolute difference between the true CATEs and their estimates (y-axis) (Section J.1). Among variants of CCN, Wass-CCN, Assign-CCN, and FCCN are able to reduce the average error by over 50%.

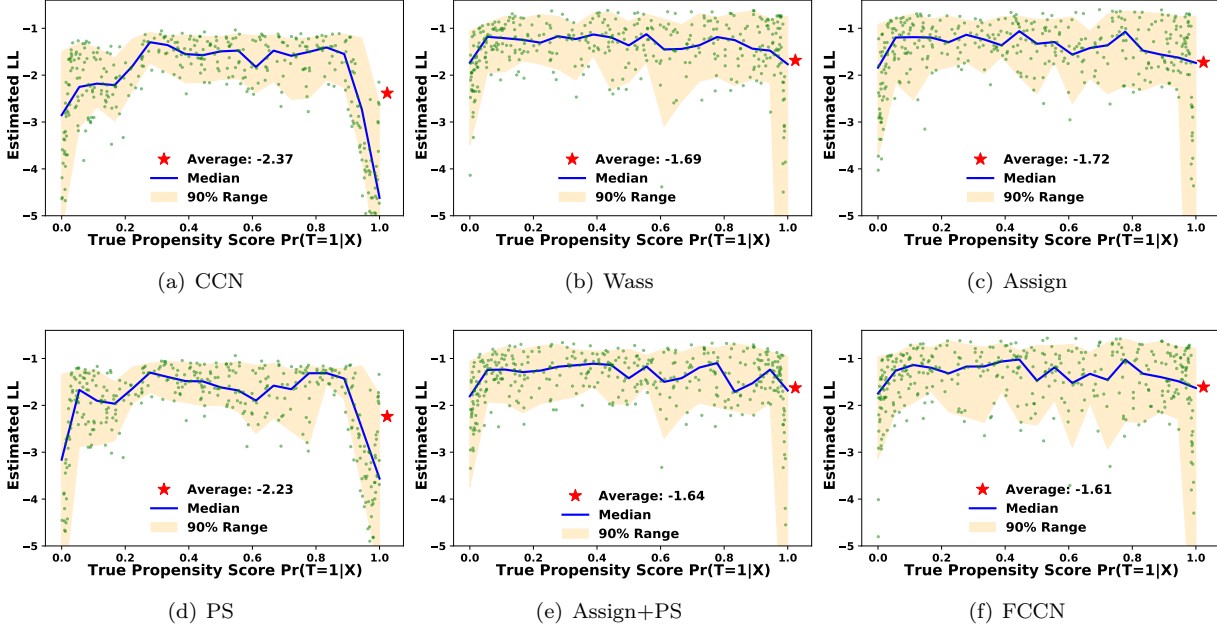

Figure S5: Scatter plot of propensity scores (x-axis) versus the estimated log-likelihood (LL) (y-axis) (Section J.1). Collectively, the Assign-loss and Wass-loss contribute to FCCN making more robust distributions estimates than any one single component.

