# OpenReview forum: "Estimating Potential Outcome Distributions with Collaborating Causal Networks"
_TMLR — Accepted by TMLR_

### Review · Reviewer_YQTK · 2022-07-16

**Summary Of Contributions:**

The paper studies estimation of potential outcome distributions using a framework dubbed Collaborating Causal Networks (CCN), extending the existing framework Collaborating Networks (CN). The problem is formalised as learning potential outcome CDFs which generalize from treatment groups to the full study  population, under assumptions often made in observational causal estimation: consistency, overlap and ignorability. It is argued that, in this setting, the asymptotic properties of CN apply also to CCN. Additionally, three heuristic adjustments to the learning objective for improving finite-sample performance of neural network estimators are proposed, grounded in the causal inference literature: (i) minimisation of representation space treatment group imbalance, (ii) propensity stratification through treatment assignment loss (iii) inclusion of fitted propensity score in representation. The resulting estimator is dubbed FCCN. The proposed framework is evaluated in two simulated datasets, IHDP and EDU, both based on randomized experiments but with synthesised outcomes. It is shown that the FCCN  adjustments to the objective of CCN  improves likelihood, error in estimated causal effects and accuracy in the estimated utility. The frameworks also compare favourably to other estimators of potential outcome distributions, including both parametric and non-parametric methods. On the EDU dataset, the authors inspect estimates more closely, and conclude that FCCN and CNN more closely fit the CDFs than baseline methods and learn at a higher rate. An ablation study concludes that each of the added components contribute to the results.

**Requested Changes:**

**Necessary for recommending acceptance**
- Adding a paragraph discussing the relation to quantile regression using neural networks and to off-policy evaluation and distributional reinforcement learning---in particular work which estimate the CDF of the value function or uses quantile regression---would be necessary to better contextualise the current work.

- Describe the architectures used to fit the CCN and FCCN models in equations, including representation components. Consider adding an illustration of CCN as well. If the CCN model does not share architecture with FCCN (but without the same objective), it would be interesting for the ablation study in should include FCCN trained only with the g-loss. In Shalit (2017), it was found that representation sharing between potential outcome estimators made the biggest improvement to estimates (not changes to the learning objective).

- Define what the PS abbreviation in the ablation study means already in 3.3.2 and describe what it does when the Assign-loss is not used.

- Include description of hyper parameter selection in main paper and a short discussion of why this method of selection is appropriate here.

- In Figure 1, g0 and g1 appear to be functions of S, but in 3.2, they are described as functions of Z and X. In other words, in (5), it is not clear how g0 and g1 relate to S.

**Would strengthen the work**
- In the EDU experiments, it is not clear whether gains over baselines come from improvements in fitting the means (which are neural networks also in the true model) or the noise distribution. Given that the PEHE improvement is quite large compared to most baselines, I would assume that a large part is due to the mean.   This merits discussion. Additionally, please state whether the "s" variable is included in "x".

- If the utility functions in 2.2. are established categories, consider adding a reference for this. Consider restructuring section 2 to better highlight what is the main problem studied in this work, what is motivation, etc.

- Consider restructuring 3 to give the reader a better idea of why we end up with the loss in (4). For example, for a fixed z, this is quantile regression, and for the expectation, you optimise the loss w.r.t. every quantile (like Zhou et al., 2021).

**Typos, minor corrections**
- p,1, abstract: "Standard causal inference assumptions". Causal inference is a large subject, there are multiple sets of standard assumptions depending on the problem studied.
- p.1, abstract: "We demonstrate CCN" -> "We demonstrate that CCN"
- p.2, "Empirically, these techniques often impose". The assumptions aren't tied to empirical observation.
- p.2, start of 2.1. "We define the covariates as X \in \cX...". This is not a definition per se, moreover X are random variables on \cX, not elements of it, no?
- p.2 "We adopt this rigor in definition". I'm not sure what this is referring to here.
- p.3 Start of 3. "Distributions Y(0)|X". Previously, this was denoted "p(Y(0)|X)"
- p.4 "Zhou et al, (2021) shows that" -> "... show that"
- p.4 "ground truth CDF functions for T \in {0,1} is Lipchitz". is -> are
- p.9 "respectfully" -> "respectively"
- The Figure 4 caption doesn't say which task is considered.
- Weakening/contextualizing Claim 2 in main contributions would better represent its reliance on previous work.
- Equation (3) is also due to Zhou et al (2021), right?

**Strengths And Weaknesses:**

**Strengths**
- The studied problem is well motivated: much attention has been paid to estimating a particular parameter of potential outcome distribution---the conditional average treatment effect (CATE)---but many other functions of the distribution could be of interest.

- The overall solution to the problem is sound, and builds naturally on previous results.

- The baseline models used in the empirical evaluation are well-chosen and represent qualitatively different approaches.

- The proposed task "EDU" is a reasonable construction given the topic of the paper. However, the noise model is somewhat simplistic given that the task is to estimate the distribution.

- The empirical results are examined from several different perspectives.

**Weaknesses**

- The structure of the paper could be improved. At a high level, the paper is about estimating p(Y(t) | X).  2.2 places large emphasis on utility functions which are only a small part of the rest of the paper. The title and Section 3 places large emphasis on extending the CN model. The main problem gets lost at times. The overall approach could be described as quantile regression with sampled target quantiles using deep neural networks. Additional losses are added to handle covariate shift.

- In 2.2., the authors discuss personalisation of treatment through the expected utility over the (to-be-estimated) potential outcome distributions. This task is closely related to off-policy evaluation, and in particular, to distributional off-policy evaluation. For example, Chandak et al (Neurips 2021), studied Universal Off-policy Evaluation and "take the first steps towards a universal off-policy estimator (UnO)—one that provides off-policy estimates and high-confidence bounds for any parameter of the return distribution". A discussion of off-policy evaluation is absent from the current paper. Similarly, the relationship to general quantile regression is not discussed.

- In the main contributions, the authors claim to "prove the asymptotic properties of CCN". These proofs are given in Appendix A and amount to arguing that if the conditional distribution is well-estimated asymptotically (proven in existing work), the same is true in expectation over different distributions of the conditioning variable (see proof of Proposition S3). While this is accurate, the claimed contribution is smaller than what the list in the introduction suggests because of its heavy reliance on previous work---I would argue that the proof is not new, even though the implication may be wider than what the initial work (Zhou et al., 2021) concluded. Moreover, the claim, as stated, could be misinterpreted as a claim about a practical algorithm, but no indication for how to actually find an optimiser of the g-loss is given.

- The proposed FCCN objective includes two hyper parameters, \alpha and \beta, which are selected (as described in Appendix C) based on the empirical log-likelihood. It is not clear to me why a non-zero value of either hyper-parameter should yield a higher log-likelihood since both are aimed at improving generalisation to a new distribution, not across samples from the same distribution. Choosing hyper parameters used for this purpose has been a large point of discussion around previous work (such as the cited Shalit et al. (2017)).

- The proof of Proposition S3 is very informal, ending with the sentence "We claim that the optimum remains invariant when we generalize from p(x) to p'(x)." Is this not what should have been proven at that point? Moreover, Lemma S1 and Lemma S2 are trivial consequences of Assumptions 1 and 3 and stating them as Lemmas is not necessary. (They are also not used to prove a Theorem).

- On page 5, the authors claim that including e( ) in the covariate space encourages propensity score stratification. The empirical support for this claim is rather mixed since including PS seems to have positive effect in some cases and negative in some (see Table 4 and Table 5). This brings up another question: In the "PS" ablation (not PS+Assign), is the Assign-loss omitted? In this case, PS has no reason to be representative of the propensity score.

- The CCN neural network parameterization is not completely described, in that 3.2 contains no reference to  the learned representation \phi. Does CCN use the same representation sharing as FCCN but without the adjustments to the objective of FCCN (this seems suggested by the ablation study in 5.4.4)? Or alternatively, are g0 and g1 fit completely independently of each other?

---

> ### Author Response · Authors · 2022-08-04
> **Response to Reviewer YQTK**
>
> We thank the reviewer for their thorough review. Below we have made efforts to either address or respond to each (paraphrased) requested change and communicated weakness. Major changes are highlighted in blue text in the revision. All typos/minor concerns have been fixed and are otherwise not addressed in this response.
>
> A. Weaknesses
>
> 1. *The authors claim to prove asymptotic properties. This contribution is smaller than suggested because of reliance on previous work.*
>     * We have modified Contribution 2 in the revised paper.
>
> 2. *It is not clear why a non-zero value of $\alpha$ or $\beta$ hyper-parameters should yield a higher log-likelihood.*
>     * We have modified the hyperparameter selection description for $\alpha$ and $\beta$ in Appendix C (Detailed Method Implementations), highlighted in blue text.
>
> 3. *The Proposition S3 proof is informal, ending with, "We claim that the optimum remains invariant when we generalize from $p(x)$ to $p'(x)$." Moreover, Lemma S1 and Lemma S2 are trivial consequences of Assumptions 1 and 3 and stating them as Lemmas is unnecessary.*
>     * We have removed this redundant sentence. We note that Lemma S1 and S2 are consequences of assumptions in Section A.2. They are used to show Propositions 1 and 2, but we have also noted in the revision that those could follow from Assumptions 1 and 3.
>
> 4. *In the "$S$" ablation, is the Assign-loss omitted? In this case, PS has no reason to be representative of propensity score.*
>     * PS is estimated through an additional model via a cross-entropy loss rather than through a shared representation. Thus, it is still representative of the propensity score.
>
> 5. *The CCN parameterization is not completely described. Does CCN use the same representation sharing as FCCN? Are $g_0$ and $g_1$ fit independently of each other?*
>     * A representation layer is shared by both $g_0$ and $g_1$, meaning that this representation should be predictive of \textit{both} potential outcome distributions. Please refer to comment B.2 for the changes made to detail the relation between CCN and FCCN.
>
> B. Requested Changes
>
> 1. *Discuss the relation to quantile regression, off-policy evaluation, and distributional reinforcement learning.*
>     * We have added paragraphs describing the relation of off-policy evaluation and quantile regression to Section 4 (Related Work).
>
> 2. *Describe the architectures of CCN and FCCN.*
>     * CCN and FCCN models are both trained via the g-loss only since we fix the function $f$ as a uniform distribution that "searches" over the outcome space. For CCN and FCCN, we use the same strategy as in Shalit et al. (2017) through learning a shared representation. We added a sentence to the end of Section 3.3 (Adjustment for Treatment Group Imbalance) describing differences between CCN and FCCN architectures. We refer to Appendix C for further details on implementations.
>
> 3. *Describe what "PS" does when the Assign-loss is not used.*
>     * We have added a definition of PS as corresponding to "propensity stratification" in Section 3.3.2. Additionally, we have added a sentence to the end of Section 3.3.2 (highlighted in blue) to clarify the purpose of PS in the absence of the Assign-loss.
>
> 4. *Include a description of hyper-parameter selection and a discussion of why this selection method is appropriate.*
>     * We include details of hyperparameter selection in Appendix C and have added a sentence to the Experiments section clarifying that hyperparameter selection is described in Appendix C.
>
> 5. *In Figure 2, $g_0$ and $g_1$ appear to be functions of $s$, but in 3.2, they are described as functions of $Z$ and $X$. It is not clear how $g_0$ and $g_1$ relate to $S$.*
>     * We have clarified the relation of $S$ to $X$ in the revised manuscript. $S$ is a learned and rearranged representation that constitutes the new space of $g_0$ and $g_1$. It could be understood as the condition for conditional distributions: Y(0)|$S$, Y(1)|$S$. $Z$ on the other hand, is a generator for any value in the space page 4, Equation (3). $g_0(z, S)$ is used to approximate the probability $\Pr(Y(0) < z|S)$.
>
> C. Would Strengthen Work
>
> 1. *In EDU experiments, it is not clear whether gains over baselines come from fitting the means or the noise distribution.*
>     * We have added the following sentence to Section 5.3 (EDU): "Figure 3 shows that FCCN is the only method able to recover both Gaussian \textit{and} exponential distributions with high fidelity, which we believe to contributes to its top performance in all metrics."
>
> 2. *Consider restructuring Section 2.*
>     * We have added to Section 2.2 (Utility Functions) to make clearer the relevance of utility functions.

---

### Review · Reviewer_4C4n · 2022-07-19

**Summary Of Contributions:**

The authors propose Collaborative Causal Networks (CCNs) and the extended full adjustment CCNs (FCCNs). Their idea is centred around CATE being a point-estimate, while the entire TE distribution would be more informative. Rather than estimating an expectation of the potential outcomes directly (as is the approach taken by most relevant methods in this area), the authors predict the entire potential outcome distribution using Collaborative Networks (CNs).

In their paper, the authors show how CNs can be extended as a T-Learner (2 distinct CNs, one for each treatment option), and suggest how one may build an S-Learner (where the treatment is considered a covariates in X). Their work is empirically validated on IHDP and EDU, two semi-synthetic datasets typically used for validating CATE models.

**Broader Impact Concerns:**

I see no broader impact concerns.

**Requested Changes:**

*Changes suggested above*
1. Provide some examples and motivation for utility functions in the main text.
2. Further motivate the use of CNs for this particular task.
3. Add CMGP as a benchmark for comparing distributions (e.g. in fig4)
4. Perform additional experiments on synthetic data with varying distributions (with increased/decreased complexity), i.e. provide additional tables such as tab1 and tab2, but using a new synthetic setup.

*Additional changes*
It is not very clear to me how CNs work. A more complete (algorithmic or architectural) description of CNs, either in the main text or the appendix, would help me a lot.

**Strengths And Weaknesses:**

STRENGTHS
1. I think, for the most part, the paper is well written and easy to follow. I have a few suggestions in this regard, which I detail in the “requested changes” section.
2. The problem makes sense. I agree with the authors’ observation that it may be more informative to estimate a potential outcome distribution rather than a point estimate. In typical scenarios, there is plenty of data to estimate these distributions accurately, yet estimating the distribution is typically not adopted in other methods.
3. I find the empirical validation of the accuracy of (F)CCN’s estimated distributions to be quite thorough and tend to believe their superiority.


WEAKNESSES
1. It is not entirely clear to me what purpose the utility functions serve. Could the authors provide an example of utility functions used in practice? Furthermore, the provided categories (i - iv) all seem to be based on an expectation over the same distribution? To me this doesn’t seem to be a good argument for estimating the distributions? Can’t we just map the point estimate to the components required to complete the utility function? Don’t get me wrong, I still think that estimating distributions is a useful idea, but the utility functions may not be the best argument for this.
2. It seems the adjustment from a standard CN to (F)CCN is to simply reinterpret it as a T-Learner. This is fine, but why not use more adopted distribution estimators? Perhaps a Gaussian Process (such as in Alaa & van der Schaar (2017)), or even a simple Kernel Density Estimator? Is there a unique property of CNs that make them more suitable for this task that I am missing?
3. While the set of benchmarks is quite good, I am missing the GPs used in Alaa & van der Schaar (2017). As their CMGP is purely based on GPs, they also end up estimating the same distributions as (F)CCN, could you use CMGPs the same way as you used (F)CNN?
4. IHDP and EDU are indeed standard benchmark tests for CATE estimators. However, the authors propose to estimate more than just CATE. How would (F)CCN react to more complicated distributions? Perhaps an additional benchmark test on synthetic data with varying distributions (according to some parameter you could visualise over) would be interesting.

---

> ### Author Response · Authors · 2022-08-04
> **Response to Reviewer 4C4n**
>
> We thank the reviewer for their thorough review. Below we have made efforts to either address or respond to each (paraphrased) requested change and communicated weakness. Major changes are highlighted in blue text in the revision. All typos/minor concerns have been fixed and are otherwise not addressed in this response.
>
> A. Weaknesses
>
> 1. *It is not entirely clear to me what purpose the utility functions serve. Could the authors provide an example of utility functions used in practice? Furthermore, the provided categories all seem to be based on an expectation over the same distribution? To me this doesn’t seem to be a good argument for estimating the distributions? Can’t we just map the point estimate to the components required to complete the utility function?*
>     * We have added to and rewritten Section 2.2 (Utility Functions) to make clearer the relation of utility functions to our work. First, noting the reviewer’s comment, we have added a comment that distributions may be of interest in themselves regardless of the utility. Second, utility functions provide a way to incorporate personalization (e.g., personal preference or needs) into decision making with regards to a given treatment, and allow for a clear evaluation of how much decisions can improve. As the reviewer points out, we could instead map the point estimate to estimate the outcome of the utility function rather than the original outcome. This strategy has been previously attempted in the literature through "Policy Learning," a framework that we discuss in the related work and compare to experimentally, revealing that estimating the potential outcome distributions and then the utility is a more reliable strategy. Additionally, we refer to the third paragraph in Section 5.3 (EDU) for a practical example of the intuition behind utility function design with regards to the EDU dataset.
>
> 2. *It seems the adjustment from a standard CN to (F)CCN is to simply reinterpret it as a T-Learner. This is fine, but why not use more adopted distribution estimators? Perhaps a Gaussian Process (e.g., Alaa and van der Schaar (2017)) or a Kernel Density Estimator?*
>     * CN is able to automatically adapt to different types of outcomes and distributions. Gaussian process-based methods are sometimes restricted by their Gaussian form/assumptions, especially if the goal is to estimate some asymmetrically distributed outcomes. Kernel density estimators rely more on an abundance of data points in the space and thus may suffer in situations where there is poor overlap of distributions. Also, to calculate probabilities using a kernel density estimator and sampling from it, would require additional numerical calculations, which is not as efficient as our CCN-based methods.
>
> 3. *While the set of benchmarks is quite good, I am missing the GPs used in Alaa & van der Schaar (2017).*
>     * We have added CMGP as a benchmark. See our response to Requested Change B.3.
>
> 4.. *IHDP and EDU are indeed standard benchmark tests for CATE estimators. However, the authors propose to estimate more than just CATE. How would (F)CCN react to more complicated distributions? An additional benchmark on synthetic data with varying distributions would be interesting.*
>     * We test CCN, FCCN, and benchmarks on synthetic multimodal-distributed data, described in Section 5.4.1. Figure 4 illustrates how CCN and FCCN better match this more complex synthetic data than the benchmarks.
>
> B. Requested Changes
>
> 1. *Provide some examples and motivation for utility functions in the main text.*
>     * We have added to and rewritten Section 2.2 (Utility Functions) to make clearer the relation of utility functions to our work.
>
> 2. *Further motivate the use of CNs for this particular task.*
>     * We have added the a comment (highlighted in blue) in the first paragraph of Section 3 (Collaborating Causal Networks) to further motivate the use of CN.
>
> 3. *Add CMGP as a benchmark for comparing distributions.*
>     * We have added CMGP as a benchmark for IHDP, EDU, synthetic multimodal data, and sample size convergence experiments, and included discussion of the cases in which it performs well and cases in which it does not have competitive performance to Section 5 (Experiments) of the paper.
>
> 4. *Perform additional experiments on synthetic data with varying distributions , i.e. provide additional tables but using a new synthetic setup.*
>     * We test CCN, FCCN, and benchmarks on synthetic multimodal-distributed data, described in Section 5.4.1. Figure 6 illustrates how CCN and FCCN better match this more complex synthetic data than the benchmarks.
>
> 5. *It is not very clear to me how CNs work. A more complete description of CNs would help me a lot.*
>     * We have included an expanded and more detailed description of the CN framework in Section 3.1 (Overview of Collaborating Networks). We have also added Figure 1, which depicts a diagram of the CN framework.

---

> > ### Comment · Reviewer_4C4n · 2022-08-07
> > **Thank you for your response**
> >
> > Dear Authors, thank you for your response. I have read and agree with your counterarguments/edits/additions.

---

### Review · Reviewer_fqza · 2022-07-21

**Summary Of Contributions:**

The paper proposes Collaborating Causal Networks (CCNs) to estimate the potential outcome distributions, which allows one to estimate the utility of the treatment even in case of personalized utility functions (i.e. individual-specific).

Proposition 1 and 2 show that CCNs asymptotically converge to the potential outcome distribution under a set of assumptions, including that the CDFs for X and the potential outcomes are Lipschitz continuous. The theoretical results are strongly based on (Zhou et al 2021).

Besides the asymptotic properties, the authors propose an adjustment for the finite sample method, FCCN, based on a regularization that encourages a domain invariant and domain specific embedding for X.

The paper provides an extensive experimental validation on some benchmark datasets that seem to validate the advantages of the methods. The ablation studies on the various components of the FCCN regularization provide some interesting insights (e.g. propensity score stratification being beneficial in IHDP, but not necessarily in general).

**Broader Impact Concerns:**

While there isn’t any specific broader impact concern for this work, the general estimation of potential outcomes distributions might easily be misused or misunderstood by practitioners, especially if the assumptions and limitations are not completely clear. I would suggest the authors discuss the limitations of their method in the conclusions.

**Requested Changes:**

I have a few questions and suggestions:

- In the introduction U(\gamma) is used, but it isn’t clear what \gamma is, so the statement U(\gamma)=\gamma for CATE is a bit unclear
- Personally I found the use of CCN for both plural (Collaborating Causal Networks) and singular (a Collaborating Causal Network) a bit confusing. I would instead use CCN for the singular and CCNs for the plural. Moreover, in the singular case I would use “a CNN”
- Some of the contributions could be stated more precisely, e.g. the asymptotic properties in 2. (under which conditions are they proven? What do these conditions, e.g. a Lipschitz CDF, entail?)
- Contribution 4. Is part of the motivation for contribution 1, and “practically more meaningful” when discussing personalized utilities seems a bit of an overstatement (since we might still need to elicit these utility functions, which seems even more complicated on an individual basis)
- The ITE vs CATE discussion is more of a footnote
- In Section 2.2. can you maybe provide for illustration and clarification purposes an example in which naively transforming the outcome for a threshold utility could result in a loss of information? (There are some results in the experimental section, but maybe you have a simpler example to show)
- Positivity can be more precisely state as  $0 < P(T=1|X=x) <1$ for all x such that $P(X=x) \neq 0$
- Section 3.1 was very short and I would have appreciated some more explanation or examples, since most of the work is based on CNs. For example I didn’t understand what does it mean that f(.) is a function that searches the outcome space, since it seems that it mostly constrains the function to an invertible one. What am I missing?
- Also I think it would be less confusing to call the network g(Y,X) in the first line as g(y,x) (similarly to later). I think this is what is intended also by the conditional CDF, which seems to be estimating P(Y < y(input) | X=x (input))
- Section 3.2 typo “Assumption 1 guarantees that …” $p(X|T=0) = p(X|T=1)$ …
- Proposition 1 seems sensible, but can you discuss how can one implement it in practice?
- Proposition 2, what does a Lipschitz CDF entail, e.g. no discrete Y ? Can you comment on how limiting this assumption is?
- In Section 3.3. Why do we need $\phi_A$, wouldn’t it be enough to have a propensity score estimation directly from X (so $e: \mathbb{R}^p \to \[0,1\]$?
- Equation 5 is unclear, what are $g_1, g_2$ ? I can assume from the text that there are two networks, each with a regularization term, but can you maybe expand the explanation?
- Typo in the VIllani(2008) citation

**Strengths And Weaknesses:**

** Strengths **
- I assume the TMLR audience is interested in the problem potential outcomes distribution estimation and its effect on decision making
- The proposed framework is an interesting application of previous work (Zhou et al 2021)
- To the best of my limited knowledge, the experimental evaluation seems extensive, including some interesting ablations and a comprehensive comparison with existing methods

** Weaknesses **
- The clarity and preciseness of the writing can be improved, including adding a bit more context and examples (more details in the requested changes)
- Some of the claims in terms of what the algorithm can do asymptotically should be qualified by the assumptions (Lipschitz continuous CDF) and its potential implications should be discussed

---

> ### Author Response · Authors · 2022-08-04
> **Response to Reviewer fqza**
>
> We thank the reviewer for their thorough review. Below we have made efforts to either address or respond to each (paraphrased) requested change and communicated weakness. Major changes are highlighted in blue text in the revision. All typos/minor concerns have been fixed and are otherwise not addressed in this response.
>
> A. Weaknesses
>
> 1. *The clarity of the writing can be improved.*
>     * We have revised the language and “flow” of the paper, and have made efforts to ensure consistency of notation to improve the clarity of our work.
>
> 2. *Some of the claims in terms of what the algorithm can do asymptotically should be qualified by the assumptions and its potential implications should be discussed.*
>     * We have highlighted the potential implications and limitations of our assumptions more clearly in the revision. Please see our revision of Contribution 2 in Section 1 (Introduction).
>
> B. Requested Changes
>
> 1. *In the introduction $U(\gamma)$ is used, but it isn’t clear what \gamma is, so the statement $U(\gamma) = \gamma$ for CATE is a bit unclear.*
>     * We have added a description of utility functions and $\gamma$ in the second paragraph of Section 1 (Introduction), and further description of utility functions to the beginning of Section 2.2 (Utility Functions).
>
> 2. *Personally I found the use of CCN for both plural and singular a bit confusing.*
>     * We have revised the paper to be more consistent with the usage of the acronym "CCN".
>
> 3. *Some of the contributions could be stated more precisely, e.g., the asymptotic properties in 2.*
>     * The primary necessary assumption is the smooth conditional CDF, which we have made clearer in the revision. Please see our revision of Contribution 2 in Section 1 (Introduction).
>
> 4. *Contribution 4 is part of the motivation for contribution 1, and "practically more meaningful" when discussing personalized utilities seems a bit of an overstatement. *
>     * We have added the distinction that this is true when personalized utilities are available. Eliciting these utility functions may be complicated in general. Please see our revision of Contribution 4 in Section 1 (Introduction).
>
> 5. *Can you provide for clarification purposes an example in which naively transforming the outcome for a threshold utility could result in a loss of information?.*
>     * We have expanded the example in Section 2.2 (Utility Functions) to provide a better intuition as to how naively transforming outcomes could result in a loss of information.
>
> 6. *Section 3.1 was very short and I would have appreciated some more explanation or examples, since most of the work is based on CNs.*
>     * We have included an expanded and more detailed description of the CN framework in Section 3.1 (Overview of Collaborating Networks). We have also added Figure 1, which depicts a diagram of the CN framework.
>
> 7. *Also, I think it would be less confusing to call the network $g(Y, X)$ in the first line as $g(y, x)$.*
>     * We have revised notation to be more consistent throughout the manuscript.
>
> 8. *Proposition 1 seems sensible, but can you discuss how can one implement it in practice?*
>     * Proposition 1 states the theoretical properties of CN-based approaches to show how it can adapt to different types and forms of outcome distributions, even in extensions to causal setups. These theoretical properties do not impact the practical implementation, as one does not need to change the structure of a CN model for it to asymptotically approach different distributions.
>
> 9. *Proposition 2, what does a Lipschitz CDF entail, e.g., no discrete Y?*
>     * A Lipschitz CDF entails PDF without point masses. This prohibits some distributions, such as zero-inflated probability distributions, but is mild in practice under an additive noise model. Please see our response to Requested Change B.3.
>
> 10. *In Section 3.3. Why do we need $\phi_A$, wouldn’t it be enough to have a propensity score estimation directly from $X$ (so $e:\mathbb{R}^p \rightarrow [0, 1]$)?*
>     * $\phi_A$ helps capture domain-specific information, which is learned/enforced through being predictive of the treatment assignment. Additionally, $\phi_A$ parameters are learned jointly with the potential outcome model. If we were to separate these out and estimate the propensity directly from $X$, then this would not be as effective a method of information sharing and we would expect model performance to suffer.
>
> 11. *Equation 5 is unclear, what are $g_0$, $g_1$?*
>     * We have added more descriptive explanations of functions $g_0$ and $g_1$ in the first paragraph of section 3.2 prior to Equation (4) where these functions are first introduced.
>
> C. Broader Impact Concerns
>
> 1. *The general estimation of potential outcomes distributions might easily be misused or misunderstood by practitioners.*
>     * We have added a paragraph discussing the broader impact and limitations of our work to the end of Section 6 (Discussion).

---

> > ### Comment · Reviewer_fqza · 2022-08-09
> > **Thank you for your response**
> >
> > The authors have addressed all of my concerns, and to what I can judge also the concerns of the other reviewers, and have revised substantially the paper. I'm quite happy with the rebuttal.

---

### Decision · Action_Editors · 2022-08-16

**Recommendation:** Accept as is

**Comment:**


The authors have substantially revised the paper and well addressed the reviewers' comments and concerns. The reviewers and I are happy to have the revised paper accepted.

With the revision, some minor textual errors have crept in that the authors should address in the camera-ready version:

1. In the new paragraph on off-policy evaluation in Sec 4, it seems like textual citations (\citet) are sometimes used when parenthetical citations (\citep) would seem better choices.

2. Something seems to have gone awry in the formulation of the first sentence in the second paragraph of the discussion section:

    "CCN and its variants are competitive with but do not best methods that ..."